# Interactions Exhibit Clustering Rhythm: A Prevalent Observation for Advancing Temporal Link Prediction

## Abstract

Temporal link prediction aims to forecast future link existence in temporal graphs, with numerous real-world applications. Existing methods often rely on designing complex model architectures to parameterize the interaction patterns between nodes. Instead, we re-think the interaction dynamics in temporal graphs (which we call "interaction rhythms") by addressing a fundamental research question: *Is there a strong yet prevalent latent interaction rhythm pattern across different temporal graphs that can be leveraged for temporal link prediction?* Our introduced empirical analyses reveal that there indeed exists temporal clustering in node interaction rhythms, where for a specific node, interactions tend to occur in bursts. Such observation leads to two key insights for predicting future links: (i) recent historical links that carry the latest rhythm pattern information; and (ii) the inter-event times that further illuminate temporal dynamics. Building on these empirical findings, we propose TG-Mixer, a novel method that explicitly captures temporal clustering patterns to advance temporal link prediction. TG-Mixer samples the most recent historical links to extract surrounding neighborhoods, preserving currently invaluable interaction rhythms while avoiding massive computations. Additionally, it integrates a carefully designed silence decay mechanism that penalizes nodes' long-term inactivity, effectively incorporating temporal clustering information for future link prediction. Both components ensure concise implementations, leading to a lightweight architecture. Exhaustive experiments on seven benchmarks against nine baselines demonstrate that TG-Mixer achieves state-of-the-art performance with faster convergence, stronger generalization capabilities, and higher efficiency. The experimental results also highlight the importance of explicitly considering temporal clustering for temporal link prediction.

## 1 Introduction

Temporal graphs can model the dynamic graph-structured data in many real-world scenarios, where objects are represented as nodes and timestamped interactions between them are depicted as temporal links Wang et al. (2021d); Tian et al. (2024a). To effectively capture the dynamic nature of such graphs, researchers have developed Temporal Graph Networks (TGNs) Souza et al. (2022); Chen & Ying (2024). These networks effectively explore the temporal and topological information inside temporal graphs, thereby facilitating representation learning. Various downstream tasks have been studied based on existing TGNs already Li et al. (2023); Su et al. (2024); Zhang et al. (2024b). Among them, temporal link prediction, aiming to forecast the future link existence between potential interaction node pairs Tian et al. (2024b), has extensive applications in various real-world systems, such as forecasting users' purchasing actions of items on recommender systems to enhance user experiences Yin et al. (2023); Zhao et al. (2024), and predicting the potential transaction between two parties on payment platforms to prevent money laundering activities Duan et al. (2024).

Existing TGNs always focus on exploring various model architectures to parameterize the interaction patterns in temporal graphs, like stacking temporal graph convolutions Zhang et al. (2023; 2024a), encoding temporal random walks Wang et al. (2021c); Jin et al. (2022), or applying sequential models Tian et al. (2024b). Although powerful, existing TGNs often invest considerable effort in approximating their parameterized interaction patterns through extensive computations. However, such procedure tends to overlook the potentially heuristic, realistic patterns inherent in the temporal

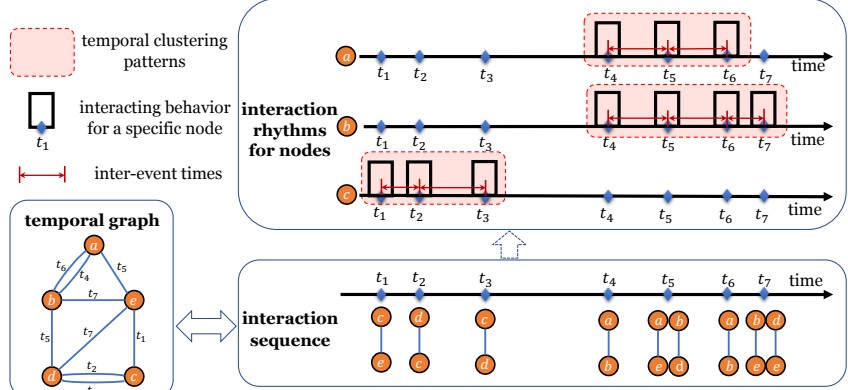

Figure 1: [New Figure] Schematic illustration for temporal clustering patterns. The behaviors of a specific node in temporal graphs exhibit burstiness in temporal dimensions.

correlations between interactions. As a result, they fail to effectively capture the invaluable patterns of interaction dynamics (we name them "interaction rhythms") shown in Figure 1, unconsciously leading the models to depend increasingly on more complex architectures to fit the fragmented details for performance improvement. Different from these existing TGNs, in this paper, we ask: *Is there a strong yet prevalent latent interaction rhythm pattern across different temporal graphs that can be leveraged for temporal link prediction?* We attempt to answer this question through the following two key contributions of our paper:

**We observe a prevalent and strong pattern of temporal clustering in node interaction rhythms.** Intuitively, the interactions of a specific node in temporal graphs present clustered occurrences in temporal dimensions over a relatively long duration. For example, a user may frequently purchase items during sales events or holidays while exhibiting few activities in other periods. Such behaviors cause the interactions of this user to occur concentratedly within specific periods, leading to temporal clustering in the interaction rhythms. To clearly illustrate temporal clustering, we introduce statistical-based empirical analyses among real-world temporal graphs from different domains. According to both macro-level analyses across the entire timeline and micro-level analyses over individual time steps (to be introduced later), the interactions of nodes tend to consistently occur in bursts. This reveals that strong temporal clustering indeed exists in temporal graphs. Inspired by such an interesting phenomenon, we explore *explicitly* incorporating temporal clustering information into temporal link prediction, achieving both effectiveness and efficiency with an elegant model design.

**We propose a novel method that captures temporal clustering for temporal link prediction.**

In this paper, we propose **TG-Mixer**, a solution that explicitly derives temporal clustering information for prediction using two primary techniques. Firstly, TG-Mixer is designed with a neighbor selection strategy that samples nodes' most recent historical links, preserving the latest invaluable rhythm patterns within the extracted neighborhoods. Secondly, TG-Mixer incorporates a temporal mixer, where we propose a silence decay mechanism to fully explore and encode temporal clustering. Specifically, for each timestamp, we introduce a novel rhythm vector that reflects the condensed essence of node interaction rhythms alongside the temporal dimensions. This rhythm vector will be decayed by penalizing the long-term inactivity of nodes, making the model effectively benefit from temporal clustering with local interaction rhythm proximities. Both components ensure a straightforward implementation and rapid computation. Consequently, as shown in Figure 2, TG-Mixer achieves outstanding performance through a lightweight model architecture that is both time-efficient and resource-effective, enabling better temporal link prediction.

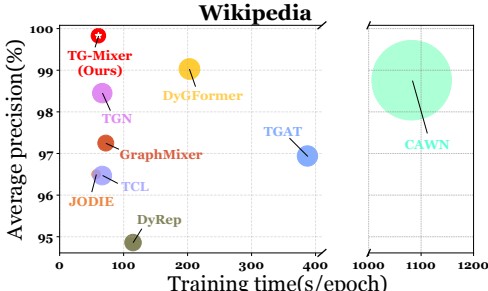

Figure 2: Comparisons of model performance, training time per epoch, number of model parameters on the Wikipedia Kumar et al. (2019) dataset. The size of each dot is proportional to the number of model parameters. TG-Mixer wins in better performance, shorter training time, and lower computational overheads. For more details, please refer to Section 5.2.

To investigate model performance, we conduct extensive experiments on seven benchmark datasets, comparing against nine baselines for temporal link prediction in both transductive and inductive settings. From the results, we find that: (i) TG-Mixer outperforms existing TGNs across all datasets with exceptionally faster convergence, stronger generalization capabilities, and higher efficiency, demonstrating the effectiveness and superiority of TG-Mixer. (ii) Comprehensive experimental results highlight the significant benefits of explicitly capturing temporal clustering, which motivates future research to re-think its importance for temporal link prediction. We also present an in-depth ablation analysis of model designs, providing a thorough understanding of TG-Mixer.

## 2 Preliminaries

In this section, we first define the temporal link prediction task and then provide the definitions of the key terminologies introduced in this paper.

### 2.1 Problem Definition

If the interactions (or links) of a graph are associated with timestamps, we name it a temporal graph.

**Definition 1.** *Temporal Graph. Given node set $\mathcal{V}$, a temporal graph defined based on it can be represented as a sequence of chronological temporal links $\mathcal{G} = \{(u_1, v_1, t_1), (u_2, v_2, t_2), ...\}$ where $0 \leq t_1 \leq t_2 \leq ....$ Each link $(u, v, t) \in \mathcal{G}$ corresponds to an interaction between a pair of interaction nodes $u \in \mathcal{V}$ and $v \in \mathcal{V}$ at timestamp $t$. Each node $u \in \mathcal{V}$ involves node feature $\boldsymbol{x}_u \in \mathbb{R}^{d_N}$ and each link $(u, v, t)$ attaches link feature $\boldsymbol{e}_{uv}^t \in \mathbb{R}^{d_L}$, where $d_N$ and $d_L$ are the feature dimensions of the nodes and links, respectively. If the graph is non-attributed, we simply let both of the node and link features to zero vectors, i.e., $\boldsymbol{x}_* = \boldsymbol{0}$ and $\boldsymbol{e}_{**}^t = \boldsymbol{0}$.*

We define temporal link prediction on the batch scale.

**Definition 2.** *Temporal Link Prediction. Given a batch size $B \in \mathbb{Z}^+$, a set of interaction nodes $\{u_b \in \mathcal{V}\}_{b=1}^B$ and $\{v_b \in \mathcal{V}\}_{b=1}^B$, and corresponding timestamps $\{t_b > 0\}_{b=1}^B$, temporal link prediction aims to predict whether each node pair $(u_b, v_b)$ interacts at timestamp $t_b$ based on the historical links $\{(u', v', t')|t' < t_b\} \subseteq \mathcal{G}$, i.e., predicting the existence of the future link $(u_b, v_b, t_b)$.*

### 2.2 Key Terminologies

Now, we define the introduced concept of node interaction rhythms among temporal graphs.

**Definition 3.** *Node Interaction Rhythm. Node interaction rhythms indicate the interaction dynamics for a specific node in temporal graphs. Given a temporal graph $\mathcal{G}$, for node $u \in \mathcal{V}$, we can derive a sequence of $u$'s interaction timestamps $\mathcal{T}_u = \{t_1', t_2'...\}$ where $0 \leq t_1' \leq t_2' \leq ....$ Each $t_i' \in \mathcal{T}_u$ records "when" the node $u$ is involved in an interaction. The distribution patterns among $\mathcal{T}_u$ encapsulate the dynamics of $u$'s interactions, which we refer to as the interaction rhythms of node $u$.*

Temporal clustering investigates the concentration pattern of node interaction rhythms.

**Definition 4.** *Temporal Clustering. Temporal clustering in node interaction rhythms reveals the following two interaction patterns among temporal graphs: (i) a specific node's interactions tend to occur frequently within certain time periods; and (ii) its interactions at other times are relatively few. This results in dense interaction rhythms during specific intervals while remaining sparse at other times, leading to a clustered pattern in the temporal dimensions. Therefore, we refer to such interaction patterns in temporal graphs as temporal clustering.*

## 3 Empirical Analyses for Temporal Clustering

To provide a straightforward illustration of temporal clustering, we carry out extensive empirical analyses to investigate temporal clustering among various real-world temporal graphs at both the macro-level (across the entire timeline) and the micro-level (over individual time steps). Specifically, we statistically quantify the temporal clustering patterns between truly existing and randomly sampled interaction nodes, analyzing the differences between these real-world and randomly constructed interaction dynamics. Finally, we try to seek some potential insights for temporal link prediction.

Given a temporal graph $\mathcal{G}$, we define all temporal links $(u_i, v_i, t_i) \in \mathcal{G}$ as positive links (the links truly exist), where the interaction nodes $u_i$ and $v_i$ represent a pair of **truly existing interaction nodes**. On the other hand, for each link $(u_i, v_i, t_i) \in \mathcal{G}$, we construct a negative link (the link that does not

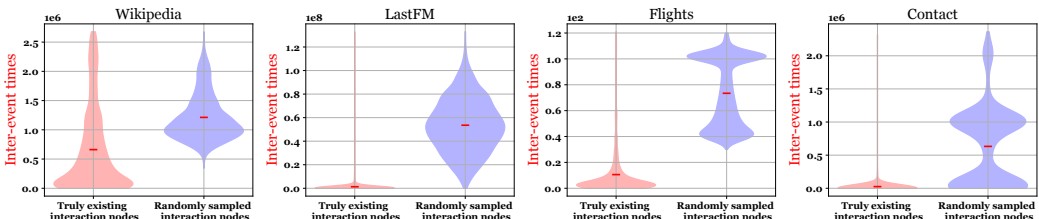

(a) Macro-level distribution of inter-event times for interaction nodes across the entire timeline: Compared to the randomly sampled interaction nodes, truly existing interaction nodes demonstrate shorter periods of inactivity and interaction bursts.

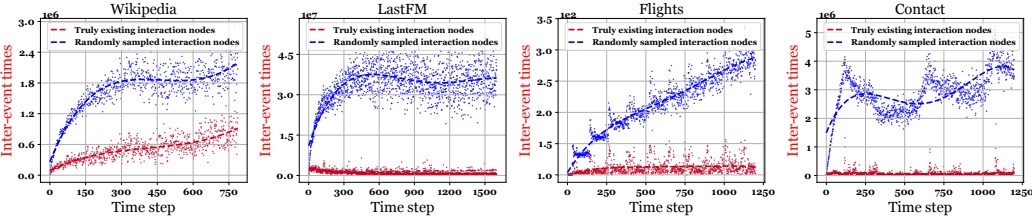

(b) Micro-level distribution of inter-event times for interaction nodes over individual time steps: Short-term interaction bursts from truly existing interaction nodes consistently occur along the temporal dimensions.

Figure 3: Both complementary empirical analyses confirm that the interactions of nodes in real-world temporal graphs exhibit prevalent and strong temporal clustering patterns. Thus, a lightweight model that explicitly considers temporal clustering information, e.g., TG-Mixer, could achieve exceptional link prediction performance with high efficiency. Analyses for other datasets are presented in Figure 9.

occur) by substituting the original interaction nodes with randomly sampled nodes, $\hat{u}_i$ and $\hat{v}_i$, thus resulting in $(\hat{u}_i, \hat{v}_i, t_i)$. We refer to $\hat{u}_i$ and $\hat{v}_i$ as a pair of ***randomly sampled interaction nodes***. To effectively quantify temporal clustering, we follow Karsai et al. (2018) and introduce ***inter-event times***. Inter-event times is defined as the time elapsed between the current timestamp and the node's last interaction timestamp. Formally, for node $u$ at timestamp $t$, we compute $u$'s inter-event times by $s_u^t = t - t'_u$ where $t'_u$ represents $u$'s last interaction timestamp prior to $t$. It is intuitive that the interactions of a node exhibiting strong temporal clustering tend to experience lower inter-event times.

**Macro-level analyses.** Macro-level analyses are conducted on the node scale. For each node $u \in \mathcal{V}$, we obtain a sequence of its interaction timestamps $\mathcal{T}_u = \{t'_1, t'_2, ..., t'_{N_u}\}$, which reveals $u$'s interaction rhythms across the entire timeline. The $N_u \in \mathbb{N}^+$ denotes the total number of interactions for node $u$ and $t'_1 \le t'_2 \le ... \le t'_{N_u}$. We compute the inter-event times for node $u$ at each interaction timestamp $t'_i \in \mathcal{T}_u$ by $s_u^{t'_i} = t'_i - t'_{i-1}$ for $i > 1$, and set $s_u^{t'_i} = t'_1$ when $i = 1$. Finally, we average the inter-event times for all the interaction timestamps of node $u$ using $\bar{s}(u) = \frac{1}{N_u} \sum_{t'_i \in \mathcal{T}_u} s_u^{t'_i}$, and then analyze the distribution of all interaction nodes using Violin Plots Hintze & Nelson (1998).

As depicted in Figure 3(a), significant differences are evident across various temporal graphs between truly existing and randomly sampled interaction nodes: Most truly existing interaction nodes involve a smaller inter-event times, indicating shorter periods of inactivity and burst-like interaction rhythms. Such characteristics strongly confirm the presence of temporal clustering patterns in temporal graphs. Conversely, the randomly constructed interaction patterns significantly reduce the likelihood of recent burst interactions, resulting in a relatively larger inter-event times. Macro-level analyses capture node interaction rhythms across the entire timeline. However, they do not directly demonstrate the consistency of this phenomenon over individual time steps. Therefore, we conduct:

**Micro-level analyses.** Micro-level analyses are performed on the timestamp scale. For each timestamp $t$, we obtain a tuple of interaction nodes[1] $\mathcal{V}_t = (u'_1, u'_2, \ldots, u'_{N_t})$, where $N_t \in \mathbb{N}^+$ is the total number of interaction nodes at timestamp $t$. We then compute the inter-event times for each node $u'_i \in \mathcal{V}_t$ at timestamp $t$ by $s_{u'_i}^t = t - t'_{u'_i}$, where $t'_{u'_i}$ denotes the last interaction timestamp of node $u'_i$ prior to $t$. Finally, we average the inter-event times for all the interaction nodes at timestamp $t$ using $\bar{s}(t) = \frac{1}{N_t} \sum_{u'_i \in \mathcal{V}_t} s_{u'_i}^t$, and show their statistical dynamics over all timestamps in Figure 3(b).

---

[1]For simplicity, we uniformly represent both the source interaction node $u/\hat{u}$ and the target interaction node $v/\hat{v}$ as $u'$.

Figure 4: Architecture of the proposed TG-Mixer. TG-Mixer explicitly considers temporal clustering by: (i) sampling the most recent historical links to preserve the latest invaluable interaction rhythms; and (ii) employing a silence decay mechanism to penalize nodes' long-term inactivity, effectively capturing temporal clustering signals for temporal link prediction.

These more fine-grained empirical analyses continue to highlight extreme differences between the two groups of interaction nodes: At most time steps, truly existing interaction nodes consistently exhibit shorter periods of inactivity and short-term interaction bursts, while randomly sampled interaction nodes typically display larger and more variable interaction patterns over time steps. This observation indicates that temporal clustering indeed exists at each time step in temporal graphs, demonstrating the prevalence and strength of temporal clustering along the temporal dimensions.

**Implications.** Such interesting interaction patterns motivate us to leverage temporal clustering as a key factor for temporal link prediction, which can be explained as follows: (i) if a given node has recent interactions, then it is more likely to interact at the current timestamp; and (ii) if the node has only distant past interactions or even no historical interactions, it is less likely to interact now.

As a result, this leads to two key insights: (i) *recent historical links*, carrying the latest rhythm pattern information, are crucial for predicting future links; and (ii) *the inter-event times*, incorporating the powerful node interaction rhythms, promotes the inclusion of temporal clustering information in predictions. Both insights require straightforward implementation and rapid computation. To this end, we could develop a lightweight model that explicitly considers temporal clustering, facilitating effective and efficient temporal link prediction.

## 4 TG-MIXER: A SOLUTION FOR CAPTURING TEMPORAL CLUSTERING

The architecture of the TG-Mixer is depicted in Figure 4. Given node $u$ at timestamp $t$, we first sample its most recent historical links to preserve fresh interaction rhythm patterns and obtain link sequences $\mathcal{S}_u^t$. Then, we encode and pad the information of those links in $\mathcal{S}_u^t$, resulting in three unified encodings. Furthermore, we concatenate these encodings and feed them into the TG-Mixer encoder, where we introduce a silence decay mechanism for fully capturing temporal clustering information. Finally, our temporal link decoder derives the temporal node representations for temporal link prediction.

**Sampling the most recent historical links.** Most existing TGNs employ neighbor selection strategies to extract nodes' surrounding neighborhoods for representation, such as "sampling multi-hop most recent links" or "extracting all 1-hop links". For example, models like TGN Rossi et al. (2020a) sample the multi-hop most recent historical links while other models like DyGFormer Yu et al. (2023) extract all 1-hop historical links. Although both strategies could help preserve temporal clustering patterns among neighborhoods, they may risk incorporating spurious or noisy information either from high-order connections Rossi et al. (2020b) or long-outdated histories Zhang et al. (2023). Besides, they could suffer from high inefficiency and complexity due to handling a massive number of selected historical neighbors Cong et al. (2023). To address these issues, we exclusively sample the most recent 1-hop historical links, preserving the latest relevant temporal clustering patterns while ensuring a conceptually and technically efficient neighbor selection strategy. Formally, for

node $u$ at timestamp $t$, we collect the sequences involving its 1-hop historical links before $t$, which is denoted as $\{(u, u', t') | t' < t\} \cup \{(u', u, t') | t' < t\}$. We only keep the top $m$ most recent historical links to obtain the sampled link sequences and denote it as $\mathcal{S}_u^t$. The number $m \in \mathbb{N}$ is a pre-defined hyper-parameter, which is analyzed in Section 5.4.

**Encoding and padding historical link information.** Given the sampled historical links $\mathcal{S}_u^t$, we first retrieve the neighbor features and link features, representing them as $\boldsymbol{X}_{u,N}^t \in \mathbb{R}^{|\mathcal{S}_u^t| \times d_N}$ and $\boldsymbol{X}_{u,L}^t \in \mathbb{R}^{|\mathcal{S}_u^t| \times d_L}$, respectively. Then, we follow Xu et al. (2020) and encode the time interval $\Delta t' = t - t'$ with a trainable periodic function to provide distinguishable temporal information from historical links, which is denoted as $\boldsymbol{X}_{u,T}^t \in \mathbb{R}^{|\mathcal{S}_u^t| \times d_T}$ where $d_T$ is the vector dimension. To capture the neighbor frequency, we apply padding if $|\mathcal{S}_u^t| < m$ and result in $\boldsymbol{P}_{u,N}^t \in \mathbb{R}^{m \times d_N}, \boldsymbol{P}_{u,L}^t \in \mathbb{R}^{m \times d_L}$, and $\boldsymbol{P}_{u,T}^t \in \mathbb{R}^{m \times d_T}$, respectively. Finally, we unify these padded features, mapping them to the same dimension $d$ with projection layers as follows:

$$\boldsymbol{Z}_{u,*}^t = \boldsymbol{P}_{u,*}^t \boldsymbol{W}_* + \boldsymbol{b}_* \in \mathbb{R}^{m \times d}, \tag{1}$$

where $\boldsymbol{W}_* \in \mathbb{R}^{d_* \times d}$ and $\boldsymbol{b}_* \in \mathbb{R}^d$ are learnable parameters, and $*$ represents $N$, $L$, or $T$. Finally, we construct the neighborhood information for node $u$ at timestamp $t$ by concatenating the unified encodings as $\boldsymbol{Z}_u^t = \boldsymbol{Z}_{u,N}^t \| \boldsymbol{Z}_{u,L}^t \| \boldsymbol{Z}_{u,T}^t \in \mathbb{R}^{m \times 3d}$, which serves as the input of TG-Mixer encoder.

**TG-Mixer encoder.** TG-Mixer is designed with (i) a token mixer that enriches the historical interaction information among the constructed neighborhoods; and (ii) a temporal mixer that explicitly incorporates temporal clustering information for representation generation. Now, we introduce these two components, respectively.

*(i) Token Mixer.* To summarize the temporal and structural information within neighborhoods, we follow Tolstikhin et al. (2021) and apply a token mixer. Specifically, we utilize a Layer Normalization (LN) layer and a Feed-Forward Network (FFN) with residual connections as follows:

$$\boldsymbol{Z}_{u,\text{token}}^t = \boldsymbol{Z}_u^t + \boldsymbol{W}_{\text{token}}^{(2)} \text{GeLU} \left( \boldsymbol{W}_{\text{token}}^{(1)} \text{LayerNorm} \left( \boldsymbol{Z}_u^t \right) \right), \tag{2}$$

where $\boldsymbol{W}_{\text{token}}^{(*)} \in \mathbb{R}^{m \times d}$ are learnable parameters. We believe this component is necessary because the token mixer enables TG-Mixer to maintain its performance by obtaining basic information from the historical interactions within neighborhoods. Consequently, even in datasets with low levels of temporal clustering, TG-Mixer could distinguish different historical link information effectively. This is empirically analyzed in Section C.13 of the Appendix.

*(ii) Temporal Mixer.* We propose a temporal mixer that explicitly models temporal clustering patterns for temporal link prediction. Specifically, at timestamp $t$, we maintain a novel rhythm vector $\boldsymbol{C}_{\text{rhythm}}^t \in \mathbb{R}^{m \times 3d}$ to encapsulate the condensed essence of historical interaction rhythms. This vector is shared communally across all nodes and will be updated globally and chronologically. As clearly illustrated in Figure 4, given a node at timestamp $t$, our temporal mixer fulfills two objectives: (i) updating the rhythm vector for the following timestamp $t'$, and (ii) producing representations that integrate temporal clustering information. Intuitively, the interaction rhythm for the next timestamp is influenced not only by the current rhythm patterns (achieved by the silence decay mechanism) but also by the historical rhythms among recent interactions (achieved by the information mixer).

- *Silence decay mechanism.* Silence decay mechanism aims to update the rhythm vector using the current rhythm information. Motivated by the insights concluded in Section 3, for node $u$ at timestamp $t$, we extract the current inter-event times, i.e., $s_u^t$, which indicates periods of inactivity for node $u$ and directly reflects its fresh rhythms. Our silence decay mechanism updates the rhythm vector by penalizing the prolonged inactivity of nodes, thus capturing temporal clustering patterns in a decaying manner. The computations are as follows:

$$\boldsymbol{C}_{\text{temp}}^t = \text{Tanh} \left( \boldsymbol{C}_{\text{rhythm}}^t \boldsymbol{W}_{\text{decay}} + \boldsymbol{b}_{\text{decay}} \right) \in \mathbb{R}^{m \times 3d}, \tag{3}$$

$$\boldsymbol{C}_{u,\text{decay}}^t = \boldsymbol{C}_{\text{rhythm}}^t - \text{g} \left( s_u^t \right) \boldsymbol{C}_{\text{temp}}^t \in \mathbb{R}^{m \times 3d}. \tag{4}$$

In this part, the rhythm vector is adjusted according to the inter-event times. We consider that $\boldsymbol{C}_{\text{rhythm}}^t$ records long-term state because it is frequently updated across timestamps. To capture fresh rhythms, we first generate short-term rhythm state $\boldsymbol{C}_{\text{temp}}^t$ by a neural network. The long-term rhythms are then decayed based on the inter-event times, with the decay coefficient $\text{g}(s_u^t) = 1 - \text{Exp}\{-2 \cdot s_u^t / T_{\max}\} \in (0, 1)$. $T_{\max}$ represents the maximum value among all inter-event times.

Table 1: AP (%) results for temporal link prediction in the transductive setting. The best model performance is highlighted as %**d** and the second-best performance is denoted in %d .

| Models | Wikipedia | Reddit | LastFM | UCI | Flights | US Legis. | Contact |
|---|---|---|---|---|---|---|---|
| JODIE | $96.50 \pm 0.14$ | $98.31 \pm 0.14$ | $70.85 \pm 2.13$ | $89.43 \pm 1.09$ | $95.60 \pm 1.73$ | $75.05 \pm 1.52$ | $95.31 \pm 1.33$ |
| DyRep | $94.86 \pm 0.06$ | $98.22 \pm 0.04$ | $71.92 \pm 2.21$ | $65.14 \pm 2.30$ | $95.29 \pm 0.72$ | $75.34 \pm 0.39$ | $95.98 \pm 0.15$ |
| TGAT | $96.94 \pm 0.06$ | $98.52 \pm 0.02$ | $73.42 \pm 0.21$ | $79.63 \pm 0.70$ | $94.03 \pm 0.18$ | $68.52 \pm 3.16$ | $96.28 \pm 0.09$ |
| TGN | $98.45 \pm 0.06$ | $98.63 \pm 0.06$ | $77.07 \pm 3.97$ | $92.34 \pm 1.04$ | $97.95 \pm 0.14$ | $75.99 \pm 0.58$ | $96.89 \pm 0.56$ |
| CAWN | $98.76 \pm 0.03$ | $99.11 \pm 0.01$ | $86.99 \pm 0.06$ | $95.18 \pm 0.04$ | $98.51 \pm 0.03$ | $70.58 \pm 0.48$ | $90.26 \pm 0.28$ |
| TCL | $96.47 \pm 0.16$ | $97.53 \pm 0.00$ | $67.27 \pm 2.16$ | $89.57 \pm 1.63$ | $91.23 \pm 0.06$ | $69.59 \pm 0.48$ | $92.44 \pm 0.12$ |
| EdgeBank | $90.37 \pm 0.00$ | $94.86 \pm 0.00$ | $79.29 \pm 0.00$ | $76.20 \pm 0.00$ | $89.35 \pm 0.00$ | $58.39 \pm 0.00$ | $92.58 \pm 0.00$ |
| GraphMixer | $97.25 \pm 0.03$ | $97.31 \pm 0.01$ | $75.61 \pm 0.20$ | $93.25 \pm 0.05$ | $90.99 \pm 0.00$ | $70.74 \pm 0.46$ | $91.92 \pm 0.03$ |
| DyGFormer | $99.03 \pm 0.02$ | $99.22 \pm 0.01$ | $93.00 \pm 0.12$ | $95.79 \pm 0.17$ | $98.91 \pm 0.05$ | $71.11 \pm 0.59$ | $98.29 \pm 0.01$ |
| TG-Mixer | $\mathbf{99.83 \pm 0.04}$ | $\mathbf{99.91 \pm 0.01}$ | $\mathbf{96.78 \pm 0.05}$ | $\mathbf{96.84 \pm 0.85}$ | $\mathbf{99.59 \pm 0.01}$ | $\mathbf{99.21 \pm 0.66}$ | $\mathbf{99.41 \pm 0.30}$ |

- *Information mixer.* To update the rhythm vector with historical rhythms and produce representations that fuse temporal clustering information, we conduct the information mixer. This component is tasked with mixing the neighborhood information and temporal clustering information, thereby updating the rhythm vector with historical rhythms among recent interactions meanwhile enabling rhythm-aware representations for link prediction. To this end, we employ an LSTM-based Yu et al. (2019) information mixer. The detailed computations are:

$$C_{\text{rhythm}}^{t'}, Z_{u,\text{temporal}}^{t} = Z_{u,\text{token}}^{t} + \text{LSTM}\left(C_{u,\text{decay}}^{t}, \text{LayerNorm}\left(Z_{u,\text{token}}^{t}\right)\right), \quad (5)$$

where $C_{\text{rhythm}}^{t'} \in \mathbb{R}^{m \times 3d}$ is the rhythm vector for the following timestamp, and $Z_{u,\text{temporal}}^{t} \in \mathbb{R}^{m \times 3d}$ is the neighborhood information that is reinforced by temporal clustering.

**Temporal link decoder.** The temporal link decoder generates temporal node representations and predicts future link existence within potential interaction nodes. For node $u$ at timestamp $t$, its representation is derived by averaging the $Z_{u,\text{temporal}}^{t}$ from TG-Mixer encoder with an output layer:

$$h_u^t = \text{Mean}(Z_{u,\text{temporal}}^{t} W_{\text{out}} + b_{\text{out}}) \in \mathbb{R}^{d_O}, \quad (6)$$

where $W_{\text{out}} \in \mathbb{R}^{3d \times d_O}$ and $b_{\text{out}} \in \mathbb{R}^{d_O}$ are learnable parameters, and $d_O$ is the output dimension. Following Yu et al. (2023), given two interaction nodes $u$ and $v$ at timestamp $t$, we predict the probability of link existence between them by applying a 2-layer MLP on their concatenated temporal representations, i.e., $p_{uv}^t = \text{MLP}\left([h_u^t \| h_v^t]\right)$. Subsequently, we employ binary cross-entropy as the loss function for model optimization.

## 5 EXPERIMENTS

### 5.1 EXPERIMENT SETTINGS

**Datasets and Baselines.** We evaluate models with seven datasets from different domains that are widely used in temporal link prediction Yu et al. (2023), including Wikipedia, Reddit, LastFM, UCI, Flights, US Legis., and Contact. Due to space limitations, the details of these datasets are illustrated in Section B.1 of the Appendix. For comparisons, we select nine representative and recent existing TGNs as our baselines, including JODIE Kumar et al. (2019), DyRep Trivedi et al. (2019), TGAT Xu et al. (2020), TCL Wang et al. (2021a), CAWN Wang et al. (2021c), TGN Rossi et al. (2020a), EdgeBank Poursafaei et al. (2022), GraphMixer Cong et al. (2023), and DyGFormer Yu et al. (2023). Descriptions of these baselines are provided in Section B.2. For all experiments, we chronologically split each dataset with ratios of $70\%$, $15\%$, and $15\%$ for training, validation, and testing, respectively.

**Tasks and Metrics.** We conduct the experiments through temporal link prediction in both transductive and inductive settings introduced by the DyGLib benchmark Yu et al. (2023). For comparison, we utilize Average Precision (AP) and Area Under the Receiver Operating Characteristic Curve (AUC-ROC) as our evaluation metrics, and all results are multiplied by 100 for clearer presentation. The detailed descriptions and other experiment settings are presented in Section B.3 of the Appendix.

**Outline.** We first discuss the main empirical results by comparing TG-Mixer with baselines in Section 5.2, then highlight the benefits of explicitly considering temporal clustering for temporal link prediction in Section 5.3 and Section 5.4. We finally provide the ablation study in Section 5.5.

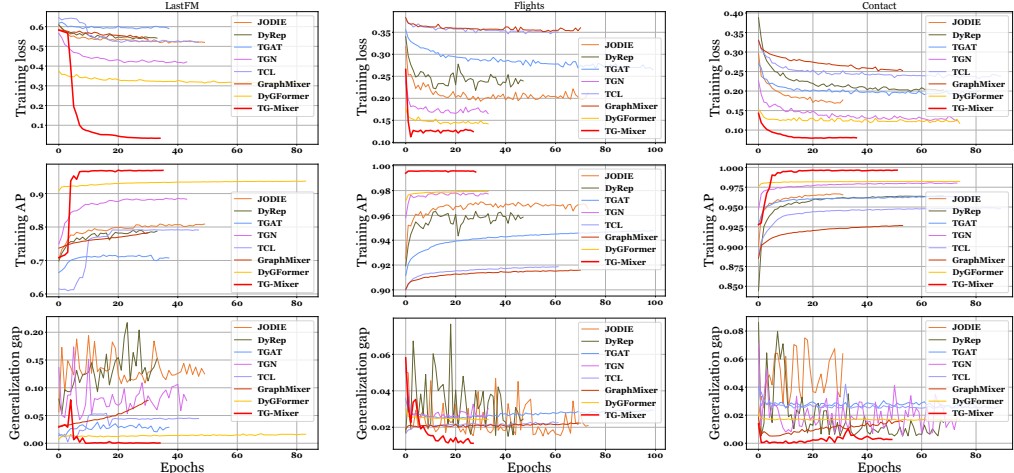

Figure 5: Comparisons of the training loss, training AP, and the generalization gap over epochs when training models. TG-Mixer demonstrates faster convergence and stronger generalization capabilities.

## 5.2 MAIN EMPIRICAL RESULTS

We start our discussions by comparing the performance and other critical capabilities between TG-Mixer and baselines, and summarize three main empirical results as follows:

**TG-Mixer achieves outstanding temporal link prediction performance.** We compare the performance with baselines in both transductive and inductive temporal link prediction tasks, presenting the AP results in Tables 1 & 8 and AUC-ROC results in Tables 9 & 10, respectively. We observe that TG-Mixer outperforms all baselines in both transductive and inductive temporal link prediction tasks across all datasets and metrics. This can be attributed to: (i) the neighbor selection strategy that samples from nodes' most recent historical links to preserve the recently invaluable interaction rhythms among extracted neighborhoods (See Section 5.4); and (ii) its ability to explicitly consider temporal clustering information to generate representations for predicting future links (See Section 5.3). Other details and discussions can be found in Sections C.5 & C.6 of the Appendix.

**TG-Mixer demonstrates faster convergence and stronger generalization capabilities.** To better analyze the model performance, we track the training loss, training AP results, and the generalization gap (the absolute gap between training and evaluation AP results) for each epoch, providing a detailed comparison of these dynamic metrics between models. The results are depicted in Figures 5 & 12. From the first two row figures, we observe that TG-Mixer consistently converges to lower training loss levels and higher training AP results within just a few epochs, demonstrating its exceptionally faster convergence. In contrast, the training curves of baselines often exhibit significant fluctuations, indicating that they could struggle to fit the fragmented interaction data during training. Interestingly, the training curves of TG-Mixer do not demonstrate distinct advantages at the start of training, where it may struggle with trivial information and suffer from the cold start issue Hao et al. (2021); Zheng et al. (2021) for capturing temporal clustering signals. After several epochs, however, it effectively leverages the powerful predictive information derived from explicitly considering temporal clustering, thus achieving enhanced performance. Additionally, the generalization gap from the third-row figures indicates the models' ability to generalize and their stability when performing on new data (the smaller the better). From the results, we find that TG-Mixer has a smaller and smoother generalization gap compared to baselines, indicating its strong generalization capabilities.

**TG-Mixer requires less training time and fewer computational overheads.** To better understand the model efficiency, we compare the training wall-clock time of a single run and the number of model parameters between TG-Mixer and baselines. The results for training time consumption and the number of model parameters are depicted in Table 2 and Table 11, respectively. Notice that the time consumption is recorded under the optimal training hyper-parameters with an early stopping strategy mentioned in Section B.3 of the Appendix. We observe that TG-Mixer requires significantly less training time compared to baselines and achieves the fastest average training speed across all datasets, demonstrating its superior efficiency. More discussions can be found in Section C.8 of the Appendix. Moreover, TG-Mixer also involves a smaller number of model parameters compared to most baselines, indicating a lower demand for computational overheads. TG-Mixer's high efficiency and low complexity demonstrate that considering temporal clustering solely with a lightweight model architecture can achieve excellent performance, validating the effectiveness of temporal clustering.

Table 2: Wall-clock training time ($\times 10^3$s) of one single run under the optimal training hyper-parameters. The results for the averaged training time of one epoch are displayed in Table 12.

| Models | Wikipedia | Reddit | LastFM | UCI | Flights | US Legis. | Contact | Avg. Rank |
|--------|-----------|--------|--------|-----|---------|-----------|---------|-----------|
| JODIE | 5.70 | 19.96 | **8.66** | 1.11 | 77.00 | **0.64** | **7.44** | 2.29 |
| DyRep | 9.86 | 26.63 | 15.20 | 1.25 | 81.42 | 1.86 | 33.73 | 4.43 |
| TGAT | 18.98 | 424.13 | 121.40 | 10.17 | 489.78 | 4.99 | 513.14 | 7.86 |
| TGN | 3.40 | 24.80 | 15.26 | 1.00 | 42.48 | 3.83 | 38.49 | 3.29 |
| CAWN | 62.76 | 212.48 | 893.54 | 33.74 | 1258.25 | 8.31 | 1088.08 | 8.86 |
| TCL | 5.79 | 37.77 | 22.87 | 1.46 | 46.58 | 2.40 | 81.61 | 4.86 |
| GraphMixer | 7.21 | 38.97 | 14.76 | 1.78 | 55.10 | 1.01 | 60.47 | 4.43 |
| DyGFormer | 9.54 | 39.28 | 874.12 | 3.80 | 409.68 | 5.48 | 291.39 | 7.14 |
| TG-Mixer | **2.18** | **10.31** | 15.76 | **0.48** | **20.32** | 0.71 | 26.85 | **1.86** |

Figure 6: Comparisons of the decay coefficients produced by our silence decay mechanism between positive and negative links. Temporal clustering can offer a highly discriminative training signal.

## 5.3 BENEFITS OF TEMPORAL CLUSTERING FROM SILENCE DECAY MECHANISM

Now, we validate the benefits of explicitly incorporating temporal clustering by our silence decay mechanism, and summarize two insightful observations as follows:

**Temporal clustering can provide a highly discriminative training signal in TG-Mixer.** To understand the effectiveness of temporal clustering, we visualize the complementary decay coefficients in Equation 9, $1 - g(s_u^t)$, from both positive and negative links during binary classification training, as described in Section B.3. A small value indicates a weak temporal clustering, resulting in more severe decay produced by the silence decay mechanism. Similar to the micro-level analyses in Section 3, we visualize the complementary coefficients between positive and negative links over individual time steps in Figures 6 & 10. Our silence decay mechanism provides a highly discriminative training signal between positive and negative links: Compared to positive links, the interaction nodes of negative links exhibit significantly lower coefficients, leading to greater decay and more severe penalties. This is because the interaction nodes of negative links tend to exhibit weaker temporal clustering, thus producing more easily distinguishable interaction rhythm signals compared to positive links. As a result, TG-Mixer can capture these dominant signals from temporal clustering, thus generating more expressive representations for prediction. This capability may be the key reason for its state-of-the-art performance with a lightweight model architecture.

**Temporal clustering can be easily adopted to boost existing sequential TGNs.** Silence decay mechanism is versatile and can be easily integrated into existing sequential TGNs. This is because these methods learn from nodes' 1-hop historical links and we can decay their neighbor information directly. The detailed implementations can be found in Section C.10 of the Appendix. We integrate the silence decay mechanism into TCL Wang et al. (2021a), GraphMixer Cong et al. (2023), and DyGFormer Yu et al. (2023), and present the temporal link prediction results in Tables 3 & 13. We find all three models demonstrate certain performance improvements after considering temporal clustering information in prediction. Additionally, the most significant performance improvements are observed in some datasets with extremely strong temporal clustering patterns revealed in our empirical analyses, such as LastFM and Flights. This observation further validates the effectiveness and importance of temporal clustering. We emphasize that TG-Mixer still outperforms, highlighting the necessity of the well-designed temporal mixer. This module allows TG-Mixer to better extract temporal clustering information compared to directly decaying neighbor information.

## 5.4 BENEFITS OF TEMPORAL CLUSTERING FROM NEIGHBOR SELECTION STRATEGY

We evaluate the neighbor selection strategy that preserves the latest invaluable temporal clustering patterns to construct neighbor information. Specifically, we compare the temporal link prediction performance with different neighbor selection strategies (both random sampling and recent sampling) under various sample sizes ($m = \{10, 20, 30, 50\}$), and present the results in Tables 4 & 14.

**The latest temporal clustering information can bring significant performance improvements.** TG-Mixer suffers from performance degradation with the random neighbor selection strategy: The

Table 3: Transductive AP (%) results of existing sequential TGNs that are boosted by temporal clustering. Before "→" are original results and after "→" are the boosting results via silence decay.

| Models | Wikipedia | Reddit | LastFM | Flights | US Legis. | Contact |
|---|---|---|---|---|---|---|
| TCL | $96.47 \to \mathbf{99.24}$ | $97.53 \to \mathbf{99.47}$ | $67.27 \to \mathbf{87.74}$ | $91.23 \to \mathbf{97.20}$ | $69.59 \to \mathbf{85.74}$ | $92.44 \to \mathbf{95.84}$ |
| GraphMixer | $97.25 \to \mathbf{99.14}$ | $97.31 \to \mathbf{98.37}$ | $75.61 \to \mathbf{95.36}$ | $90.99 \to \mathbf{92.51}$ | $70.74 \to \mathbf{96.42}$ | $91.92 \to \mathbf{97.74}$ |
| DyGFormer | $99.03 \to \mathbf{99.64}$ | $99.22 \to \mathbf{99.52}$ | $93.00 \to \mathbf{94.60}$ | $98.91 \to \mathbf{99.08}$ | $71.11 \to \mathbf{90.42}$ | $98.29 \to \mathbf{99.10}$ |
| TG-Mixer | **99.83** | **99.91** | **96.78** | **99.59** | **99.21** | **99.41** |

best performance is achieved using the most recent selection strategy, which directly reflects the latest temporal clustering patterns among sampled neighborhoods. Conversely, the random neighbor selection strategy fails to protect the complete interaction rhythm patterns, destroying the temporal clustering information from the input neighborhoods and thus leading to performance degradation. Furthermore, we also observe that different datasets show varying sensitivities to the sample size. For example, optimal results under the recent neighbor selection strategy are obtained at $m = 30$ for Wikipedia and $m = 10$ for Reddit, respectively. Therefore, it is necessary to tune the sample size $m$ based on specific datasets to find the optimal hyper-parameters for temporal link prediction. We set the optimal $m$ as the default hyper-parameter for each dataset mentioned in Section B.3.

Table 4: Transductive AP (%) results of diverse neighbor selection strategies in various sample sizes.

| Dataset | # Sample size | Recent sample | Random sample | Dataset | # Sample size | Recent sample | Random sample | Dataset | # Sample size | Recent sample | Random sample |
|---|---|---|---|---|---|---|---|---|---|---|---|
| Wikipedia | 10 | 99.26 | 93.85 | Reddit | 10 | **99.91** | 98.16 | UCI | 10 | 96.21 | 95.46 |
|  | 20 | 99.54 | 94.21 |  | 20 | 99.83 | 98.33 |  | 20 | **96.84** | 95.40 |
|  | 30 | **99.83** | **94.62** |  | 30 | 99.42 | **98.74** |  | 30 | 96.45 | **95.53** |
|  | 50 | 99.79 | 94.58 |  | 50 | 99.08 | 98.51 |  | 50 | 96.00 | 95.20 |

## 5.5 ABLATION STUDY

We also conduct an ablation study to validate the effectiveness of each component within the TG-Mixer. Detailed implementations of the variants can be found in Section C.12 of the Appendix. From the results of rows 1, 2, and 5, we find that using information mixer achieves optimal performance, while the implementation of attention mechanisms results in performance degradation. Although these complex attention mechanisms could try their best to optimize and balance all trivial information details in nodes' interactions, they fail to adequately focus on high-level temporal clustering patterns among interaction dynamics. Moreover, by comparing the results of rows 3, 4, and 5, we observe that TG-Mixer obtains the best performance when using the temporal mixer, further demonstrating the effectiveness of the temporal mixer that fully captures temporal clustering information for advancing temporal link prediction. Additionally, by comparing the results of rows 5 and 6, removing the silence decay mechanism in TG-Mixer will lead to a significant performance decrease. This demonstrates the importance of capturing temporal clustering information for temporal link prediction.

Table 5: Transductive AP (%) results of ablation study.

| Techniques | Variants | Wikipedia | Reddit | UCI | Contact |
|---|---|---|---|---|---|
| Attention mechanism | Full attention Vaswani et al. (2017) | 97.30 | 97.82 | 77.45 | 92.94 |
|  | Temporal attention Xu et al. (2020) | 96.94 | 98.52 | 79.63 | 96.28 |
| Information mixer | MLP-Mixer Cong et al. (2023) | 97.25 | 97.31 | 93.25 | 91.92 |
|  | w/. silence decay mechanism | 99.14 | 98.37 | 95.73 | 97.74 |
|  | w/. temporal mixer (*i.e., TG-Mixer*) | **99.83** | **99.91** | **96.84** | **99.41** |
|  | TG-Mixer$_{g(\cdot)=0}$ | 97.34 | 96.10 | 93.91 | 94.81 |

## 6 CONCLUSION

In this paper, we observe a prevalent yet strong pattern of temporal clustering in node interaction rhythms among temporal graphs and utilize this interesting phenomenon for advancing temporal link prediction. We propose TG-Mixer, an effective and efficient method that explicitly considers temporal clustering information by learning from the most recent historical links and penalizing the nodes' long-term inactivity in a decaying manner. Extensive experimental results demonstrate the superiority of TG-Mixer and the benefits of temporal clustering, motivating us to re-think the importance of incorporating interaction dynamics for temporal link prediction. In the future, not limited to temporal link prediction, we will extend our algorithms for different downstream tasks, such as evolving node classification.

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

# A    THEORETICAL ANALYSIS

In this section, we provide the theoretical analysis of our proposed silence decay mechanism using the theory of Hawkes Process. Hawkes Process Brémaud & Massoulié (1996) is a stochastic process that allows the occurrence of an event to increase the probability of future events. The occurrence of an event raises the probability of a new event occurring within a short period afterward. This self-excitation makes the Hawkes Process particularly suitable for describing sequences of interactions characterized by temporal clustering. We first present the basic definition of the Hawkes Process.

**Definition 1. Hawkes Process.** For $K \in \mathbb{N}$ and $[K] = 1, \ldots, K$, the point process can be desribed as $N = (N_1^t, \cdots, N_K^t)$, where $N_k^t$ represents the number of events that have occurred until timestamp $t$ at location $k$, and $t \in \mathbb{R}^+$. Its dynamics are characterized by a conditional intensity function $\lambda^t = (\lambda_1^t, \cdots, \lambda_K^t)$, which is informally the infinitesimal rate of an event conditionally on the past of the process. In the nonlinear Hawkes model, the intensity process has the following form:

$$\lambda_k^t = \mu_k + \sum_{j=1}^{K} \int_{-\infty}^{t} \gamma_{kj}(t-s) dN_j^s, \quad t \in \mathbb{R}, \quad k, j \in [K], \tag{7}$$

where $\mu_k > 0$ denotes the background or spontaneous rate of events. The function $\gamma_{kj} : \mathbb{R}^+ \to \mathbb{R}$ is an exciting function that models the effects of past history on the current event.

In a network system, various factors may lead to an interaction. One of the most important factors describes the cascade of historical influences on the dynamic network system Crane & Sornette (2008). This process captures how previous attention from one individual's history can spread to the present time and become the cause that triggers their future attention. Therefore, if a given entity whose interests make it susceptible to a certain recent interaction, it may trigger action through a cascade of intermediate steps in the temporal dimension Demirkan et al. (2013), leading to temporal clustering effects in dynamic network systems.

**Definition 2. Temporal clustering effects.** Temporal clustering effects investigate how the distribution of waiting times Vázquez et al. (2006) (i.e., inter-event times in our paper) is modified by the combination of interactions and historical influences in a dynamic network system. Such a pattern can be conveniently modeled by the self-excited Hawkes Process. Specifically, the likelihood of an entity $u$ having an interaction at timestamp $t$ can be quantified by the conditional strength of the interaction:

$$\lambda_u^t = \mu_u^t + \sum_{t' < t} \gamma_u^{t'} \rho(t - t'), \tag{8}$$

where $\mu_u^t$ is the underlying rate of interactions occurring in $u$ at timestamp $t$, independent of historical interactions on $u$. $\gamma_u^{t'}$ denotes the amount of excitation induced by the historical interactions at $t'$ $(t' < t)$ to the current interaction, and $\rho(\cdot)$ is a kernel function capturing the time decay effects.

Interestingly, the scheme of silence decay in this paper hits the formulation of temporal clustering effects perfectly, where each node recursively receives decaying information from its inter-event times in multiple timestamps. In our paper, this process is written as follows:

$$\boldsymbol{C}_{u,\text{decay}}^t = \boldsymbol{C}_{\text{rhythm}}^t - \mathrm{g}\left(s_u^t\right) \boldsymbol{C}_{\text{temp}}^t. \tag{9}$$

**Remark.** In Equation 9, the output of the silence decay for a node is derived by receiving and superposing information from the current state of our introduced rhythm vector and the decaying information from the temporal clustering, respectively. The self-state is responsible for capturing the fundamental intensity while the decaying information captures the excitation caused by historical interaction patterns of nodes. Therefore, TG-Mixer can effectively capture the node's persistent influence from temporal clustering effects, achieving superior performance for temporal link prediction.

# B    SUPPLEMENTARY EXPERIMENTAL SETTINGS

## B.1    DETAILED DESCRIPTIONS OF DATASETS

We employ seven widely-used datasets[2] in this paper and summarize their detailed statistics in Table 6 where "# Feature" indicates the number of link feature dimensions. The detailed descriptions of these datasets are presented as follows:

---

[2]Download from `https://zenodo.org/records/7213796#.Y1cO6y8r30o`

- **Wikipedia** is a temporal graph that records the edits made to Wikipedia pages over a month. In Wikipedia, nodes correspond to users or pages and temporal links with timestamps represent editing activities. Each of the links includes a 172-dimensional feature based on the Linguistic Inquiry and Word Count (LIWC) Pennebaker et al. (2001).

- **Reddit** is a temporal graph that tracks user posts across subreddits over a month. In Reddit, nodes are users or subreddits and timestamped links are posting actions. Additionally, each link is characterized by a 172-dimensional LIWC feature.

- **LastFM** is a temporal graph that records interaction data where users listen to songs over a month. In LastFM, users and songs are represented as nodes, while links between them denote the listening activities of users.

- **UCI** is a temporal graph that captures online communication activities among students from a university. In UCI, the nodes represent students, and the timestamped links between them denote established dialogues.

- **Flights** is a temporal graph that records air traffic conditions during the COVID-19 pandemic period. In Flights, airports are modeled as nodes and the flight routes between airports are modeled as timestamped links. Each link contains a weight, representing the number of flights on the corresponding route within a day.

- **US Legis.** is a temporal graph that tracks co-sponsorships among legislators from the US Senate. In US Legis., nodes represent the legislators and links with timestamps indicate the social sponsorship interactions between them. Each link carries a weight, representing the frequency where two congresspersons have co-sponsored a bill within the same congress session.

- **Contact** is a temporal graph that describes the physical proximity among university students over four weeks. In Contact, nodes represent students and links with timestamps indicate that two nodes are within close proximity. Each link carries a weight, representing the degree of physical proximity between students.

Table 6: Detailed statistics of datasets.

| Dataset | # Nodes | # Links | # Feature | Duration | Time Steps | Domains | Time Granularity |
|---|---|---|---|---|---|---|---|
| Wikipedia | 9,227 | 157,474 | 172 | 1 month | 152,757 | Social | Unix Timestamps |
| Reddit | 10,984 | 672,447 | 172 | 1 month | 669,065 | Social | Unix Timestamps |
| LastFM | 1,980 | 1,293,103 | – | 1 month | 1,283,614 | Interaction | Unix Timestamps |
| UCI | 1,899 | 59,835 | – | 196 days | 58,911 | Social | Unix Timestamps |
| Flights | 13,169 | 1,927,145 | 1 | 4 months | 122 | Transport | days |
| US Legis. | 255 | 60,396 | 1 | 12 congresses | 12 | Politics | congresses |
| Contact | 692 | 2,426,279 | 1 | 1 month | 8,064 | Proximity | 5 minutes |

## B.2 DETAILED DESCRIPTIONS OF BASELINES

We select nine representative and recent existing TGNs for performance evaluation and capability discussions. Below are their detailed descriptions:

- **JODIE** Kumar et al. (2019) is designed to handle user-item bipartite temporal interaction graphs. It simply employs a pair of Recurrent Neural Networks Sherstinsky (2020) to update the states of users and items, respectively. Additionally, a projection layer is used to learn the trajectory of node representations, thereby mitigating the issue of outdated representations.

- **DyRep** Trivedi et al. (2019) starts to consider neighborhood information and introduces a temporal-attentive aggregation module to capture the temporally evolving structural information in nodes' neighborhoods among temporal graphs.

- **TGAT** Xu et al. (2020) proposes a temporal attention mechanism to aggregate information from temporal-topological neighbors, generating temporal node representations in temporal

graphs. It also introduces a trainable time encoding function to provide basically distinguishable temporal information, which has been widely adopted in the subsequent TGNs' architectures.

- **TGN** Rossi et al. (2020a) synthesizes the approaches of the above models and proposes a memory module that maintains a state vector for each node. TGN updates nodes' memory whenever they engage in interactions. It also introduces a message-related module, a memory update module, and a temporal embedding module to generate temporal representations for nodes among temporal graphs.

- **CAWN** Wang et al. (2021c) employs temporal walks to generate node representations. It first extracts multiple anonymous random temporal walks starting from the central node and utilizes a Recurrent Neural Network to encode them. CAWN then aggregates these temporal walks to produce the final temporal node representation for temporal link prediction.

- **EdgeBank** Poursafaei et al. (2022) is an entirely memorization-based method without any trainable parameters for transductive temporal link prediction. It utilizes a memorization module to memorize previously observed links using various strategies. A given link would be predicted as positive if it can be found in current memorization module, and as negative otherwise.

- **TCL** Wang et al. (2021a) employs a breadth-first search algorithm within its constructed temporal dependency interaction sub-graph to extract interaction sequences. It then introduces a Transformer encoder that considers both topological and temporal information to learn the representations of central nodes. TCL also developed a cross-attention mechanism in Transformer to model the inter-dependencies between two interaction nodes.

- **GraphMixer** Cong et al. (2023) incorporates a link encoder based on the MLP-Mixer Tolstikhin et al. (2021) to generate temporal node representations. It also introduces a fixed time encoding function that outperforms the traditional learnable versions among its design. Additionally, GraphMixer adopts a node encoder with mean-pooling to summarize the link information.

- **DyGFormer** Yu et al. (2023) leverages 1-hop neighbor information for temporal graph representation learning. It utilizes a Transformer encoder equipped with a patching technique to effectively capture long-term dependencies among nodes in temporal graphs. Additionally, DyGFormer integrates a Neighbor Co-occurrence Feature to retain correlation information between source and target nodes.

### B.3 IMPLEMENTATION DETAILS

**Tasks and Metrics.** We conduct the experiments through the temporal link prediction task. Following Zhang et al. (2023), we employ two settings: the transductive setting, which predicts future links between nodes that have been observed during the training phase, and the inductive setting, which involves link prediction between unseen nodes. Moreover, we use Average Precision (AP) and Area Under the Receiver Operating Characteristic Curve (AUC-ROC) as our evaluation metrics. It is important to note that we follow the same configures as Yu et al. (2023), and thus, we maintain the table numbers of baselines reported in its publication.

**Training and Evaluation.** We follow Yu et al. (2023) and adopt a mini-batch training process. Specifically, we identify the links within each mini-batch as positive and generate an equal number of negative links by randomly sampling node pairs within the training data. In practice, to improve consistency, we fix the source nodes and sample an equal number of the destination nodes to construct negative links. Therefore, the negative edges share the same source nodes as the positive edges. Subsequently, we can utilize our temporal link decoder mentioned in Section 4 for binary classification to predict the link existence among these node pairs. Additionally, recall that we maintain a rhythm vector globally for all nodes and update it along the timestamps. For batch processing, we maintain a rhythm matrix with a size corresponding to the batch size, which is dynamically updated throughout training. We train the models for 100 epochs and employ an early stopping strategy with a patience of 20. For all results, we utilize Adam Kingma & Ba (2014) for model optimization and standardize the learning rate and batch size across all models and datasets to 0.0001 and 200, respectively. To ensure reliability, we run the models five times with seeds ranging from 0 to 4 and report the averaged

performance to minimize variations. All experiments are conducted on a single server equipped with 72 cores, 32GB of memory, and an Nvidia Tesla V100 GPU.

**Model Configurations.** For all the baselines, we follow a recently popular temporal graph representation learning library named DyGLib[3] Yu et al. (2023). This library performs an exhaustive grid search to identify the optimal configurations of critical hyper-parameters and rectifies certain technical bugs among the original implementations of existing TGNs, thus typically achieving enhanced performance. As for TG-Mixer, there is only one hyper-parameter mentioned in Section 4, i.e., the sample size for neighbor selection strategy $m$ (analyzed in Section 5.4). In practice, we set $m = 30$ by default for Wikipedia, $m = 10$ for Reddit and LastFM, and $m = 20$ for the remaining datasets.

## C  Supplementary Experimental Results

### C.1  Additional Experiments on the TGB Benchmark

To fully investigate model performance, we conduct the additional evaluation on the recently introduced Temporal Graph Benchmark, TGB[4] Huang et al. (2024), which contains more challenging datasets and tasks for model evaluation. We conduct additional experiments on five datasets, including Wikipedia, Review, Coin, Comment, and Flight. For the dynamic link property prediction task, we sample 100 negative edges per positive edge and employ Mean Reciprocal Rank (MRR) as our evaluation metric. We evaluate the performance of seven baselines and keep the same model configurations used in our paper.

From the results in Table 7, we find that TG-Mixer still demonstrates outstanding or competitive performance under the MRR metrics and datasets on the TGB benchmark, further proving its effectiveness for dynamic link property prediction. Additionally, TG-Mixer ranks first on the Wikipedia dataset and third on the Coin dataset. We note that the surprise index (the ratio of test links that are not seen during training defined in Poursafaei et al. (2022)) for Wikipedia and Coin are 0.108 and 0.120, respectively, which indicates that more unseen links exist in Coin during testing. This may conflict with the motivation behind our temporal clustering design, leading to a suboptimal performance on the Coin dataset. Moreover, we notice that TG-Mixer also demonstrates a significantly superior performance on the Review, Comment, and Flight datasets, further highlighting its strong capabilities and robustness. Additionally, the most significant performance improvements are observed in the Review and Flight datasets. This can be attributed to their higher interaction density (computed as # node degree / #time steps, as described in Section D.4), which indicates that the nodes in these datasets feature a larger number of simultaneous interactions. Such interaction patterns result in pronounced burstiness and stronger temporal clustering, significantly enhancing the effectiveness of our model.

Table 7: [New Table] MRR (%) results for the dynamic link property prediction task with 100 negative edges per each positive edge. The best model performance is highlighted in %d and the second-best performance is denoted in %d . "NA" denotes scenarios where a specific method was not applied to the dataset due to computational issues.

| Models | Wikipedia | Review | Coin | Comment | Flight |
|---|---|---|---|---|---|
| DyRep | $51.91 \pm 1.95$ | $40.06 \pm 0.59$ | $45.20 \pm 4.6$ | NA | NA |
| TGAT | $59.94 \pm 1.63$ | $19.64 \pm 0.23$ | $60.92 \pm 0.57$ | $56.20 \pm 2.11$ | NA |
| TGN | $68.93 \pm 0.53$ | $37.48 \pm 0.23$ | $58.60 \pm 3.7$ | NA | NA |
| TCL | $78.11 \pm 0.20$ | $16.51 \pm 1.85$ | $68.66 \pm 0.30$ | $70.11 \pm 0.83$ | NA |
| GraphMixer | $59.75 \pm 0.39$ | $36.89 \pm 1.50$ | $\mathbf{75.57 \pm 0.27}$ | $76.17 \pm 0.17$ | $77.66 \pm 1.98$ |
| EdgeBank | $52.50 \pm 0.00$ | $2.29 \pm 0.00$ | $35.90 \pm 0.00$ | $12.85 \pm 0.00$ | $16.70 \pm 0.00$ |
| DyGFormer | $79.83 \pm 0.42$ | $22.39 \pm 1.52$ | $75.17 \pm 0.38$ | $67.03 \pm 0.14$ | NA |
| TG-Mixer | $\mathbf{80.80 \pm 0.27}$ | $\mathbf{49.92 \pm 1.01}$ | $75.09 \pm 0.29$ | $\mathbf{80.17 \pm 0.61}$ | $\mathbf{84.88 \pm 2.10}$ |

---

[3] https://github.com/yule-BUAA/DyGLib
[4] https://tgb.complexdatalab.com/

## C.2 ADDITIONAL EXPERIMENTS UNDER DIFFERENT NEGATIVE SAMPLING STRATEGIES

Inspired by recent studies Poursafaei et al. (2022); Huang et al. (2024) that provide the suggestion of robust evaluation in temporal link prediction, we adopt more challenging evaluation scenarios with different negative sampling strategies and rank-based evaluation metrics to carefully assess model performance. Based on this motivation, we conduct additional experiments to evaluate the models under random negative sampling, historical negative sampling, inductive negative sampling, and degree-aware negative sampling using the MRR metric.

In addition to the random negative sampling strategy used in the main experiments, we follow Poursafaei et al. (2022) and employ historical negative sampling (sampling negative links that have been observed before but are absent in the current step) and inductive negative sampling (sampling negative links are not observed during training). To further mitigate potential biases from popular nodes during evaluation, we construct negative links by sampling negative destination nodes based on their historical degree distribution. Specifically, we sample a destination node with the probability of $d/N$, where $d$ is the node's current degree and $N$ denotes the total number of historical interactions. For the evaluation metrics, we follow TGB Huang et al. (2024) and sample 100 negative edges per positive edge and employ Mean Reciprocal Rank (MRR) as our evaluation metric.

From the results shown in Figure 7, we find that: (i) TG-Mixer consistently demonstrates strong performance across all negative sampling strategies under the MRR metric, further proving its robustness and effectiveness for temporal link prediction. (ii) All models experience some degree of performance degradation under different negative sampling strategies. Such challenging evaluation scenarios also amplify the differentiation in model performance. (iii) Inductive negative sampling tends to cause the most performance drop. This is likely because this strategy samples unseen nodes to construct negative links, making it more challenging for models to accurately distinguish between positive and negative links.

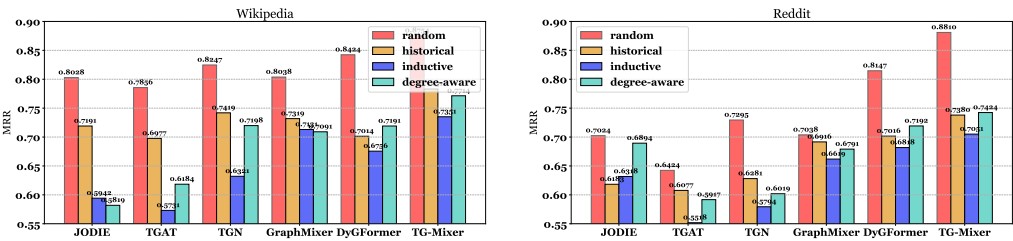

Figure 7: [New Figure] Transductive MRR results under different negative sampling strategies.

## C.3 ADDITIONAL EXPERIMENTS UNDER DIFFERENT BATCH SIZES

In this section, we evaluate model performance under different batch sizes. The batch-based training approach, introduced by Rossi et al. (2020a), has been widely adopted in the temporal graph community. This method processes interactions in batches following their chronological order. Specifically, it first sorts all interactions by timestamp and then groups them into batches using the predefined batch size. In this way, Back Propagation Through Time (BPTT) within each batch allows the models to update in chronological order. Despite its efficiency, a fundamental issue with this approach is that all predictions within a given batch share the same model state, which may be outdated for later interactions in the batch. It is particularly problematic for memory-based methods, as they always explicitly maintain an up-to-date memory component, such as the rhythm vector in TG-Mixer. While the memory state for the first interaction in the batch is up-to-date (as it incorporates information from all previous interactions), the memory state for the last interaction in the batch is out-of-date (as it does not include information from previous interactions within the same batch).

Based on the above motivation, we conduct additional experiments to evaluate model performance under different batch sizes. From the results in Figure 8, we find that: (i) TG-Mixer continues to achieve the best performance across various batch sizes, further validating its effectiveness and robustness for temporal link prediction. (ii) Although the performance ranking of models remains

unchanged, memory-based methods (e.g., TG-Mixer and TGN) tend to perform better with smaller batch sizes. This underscores the limitations of the existing batch training approach when applied to larger batch sizes. It also highlights the need for a more effective parallel processing method, which is beyond the scope of this work and we leave this as a future research direction.

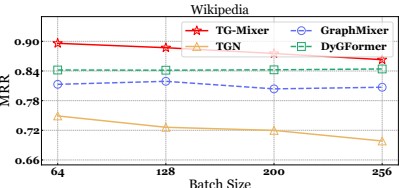 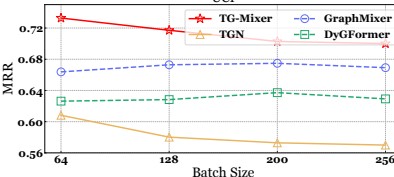

Figure 8: [New Figure] Transductive MRR results under different batch sizes.

## C.4 SUPPLEMENTARY RESULTS FOR MACRO-LEVEL AND MICRO-LEVEL ANALYSES

We present the supplementary macro-level analyses and micro-level analyses on other datasets in Figure 9. We find that temporal clustering in node interaction rhythms is a widespread and strong characteristic among various real-world temporal graphs from different domains.

## C.5 DETAILS FOR INDUCTIVE TEMPORAL LINK PREDICTION WITH AP METRIC

Inductive temporal link prediction focuses on forecasting future link existence between two unseen nodes. The inductive setting effectively prevents the model from merely memorizing previously observed node information, thus evaluating the true predictive capabilities and reliability of models. Specifically, when preparing the training data, we select 10% of nodes as inductive nodes and remove them from the training split. Subsequently, inductive temporal link prediction aims to forecast the future link existence between these inductive nodes during the evaluation and testing phases. We present the AP results for inductive temporal link prediction in Table 8. It's important to note that EdgeBank directly predicts links between unseen nodes as negative, making it unsuitable for the inductive setting. We find that TG-Mixer still performs best, demonstrating its effectiveness and crucial importance in capturing temporal clustering for better temporal link prediction.

Table 8: AP (%) results for temporal link prediction in the inductive setting. The best model performance is highlighted in %d and the second-best performance is denoted in %d . Note that EdgeBank Poursafaei et al. (2022) directly predicts links between unseen nodes as negative, and therefore, it cannot be applied to the inductive setting.

| Models | Wikipedia | Reddit | LastFM | UCI | Flights | US Legis. | Contact |
|---|---|---|---|---|---|---|---|
| JODIE | $94.82 \pm 0.20$ | $96.50 \pm 0.13$ | $81.61 \pm 3.82$ | $79.86 \pm 1.48$ | $94.74 \pm 0.37$ | $54.93 \pm 2.29$ | $94.34 \pm 1.45$ |
| DyRep | $92.43 \pm 0.37$ | $96.09 \pm 0.11$ | $83.02 \pm 1.48$ | $57.48 \pm 1.87$ | $92.88 \pm 0.73$ | $57.28 \pm 0.71$ | $92.18 \pm 0.41$ |
| TGAT | $96.22 \pm 0.07$ | $97.09 \pm 0.04$ | $78.63 \pm 0.31$ | $79.54 \pm 0.48$ | $88.73 \pm 0.33$ | $51.00 \pm 3.11$ | $95.87 \pm 0.11$ |
| TGN | $97.83 \pm 0.04$ | $97.50 \pm 0.07$ | $81.45 \pm 4.29$ | $88.12 \pm 2.05$ | $95.03 \pm 0.60$ | $58.63 \pm 0.37$ | $93.82 \pm 0.99$ |
| TCL | $96.22 \pm 0.17$ | $94.09 \pm 0.07$ | $73.53 \pm 1.66$ | $87.36 \pm 2.03$ | $83.41 \pm 0.07$ | $52.59 \pm 0.97$ | $91.11 \pm 0.12$ |
| CAWN | $98.24 \pm 0.03$ | $98.62 \pm 0.01$ | $89.42 \pm 0.07$ | $92.73 \pm 0.06$ | $97.06 \pm 0.02$ | $53.17 \pm 1.20$ | $89.55 \pm 0.30$ |
| GraphMixer | $96.65 \pm 0.02$ | $95.26 \pm 0.02$ | $82.11 \pm 0.42$ | $91.19 \pm 0.42$ | $83.03 \pm 0.05$ | $50.71 \pm 0.76$ | $90.59 \pm 0.05$ |
| DyGFormer | $98.59 \pm 0.03$ | $98.84 \pm 0.02$ | $94.23 \pm 0.09$ | $94.54 \pm 0.12$ | $97.79 \pm 0.02$ | $54.28 \pm 2.87$ | $98.03 \pm 0.02$ |
| TG-Mixer | $99.83 \pm 0.05$ | $99.89 \pm 0.01$ | $95.91 \pm 0.59$ | $96.56 \pm 0.86$ | $99.09 \pm 0.01$ | $99.21 \pm 0.86$ | $99.58 \pm 0.36$ |

## C.6 DETAILS FOR TEMPORAL LINK PREDICTION WITH ROC-AUC METRIC

ROC-AUC is also a popular evaluation metric used in temporal graph learning Wang et al. (2021d). The larger the numbers, the better the model performance. The results under both transductive and inductive settings are depicted in Tables 9 & 10, respectively. We can observe that TG-Mixer still achieves outstanding performance across all datasets in both transductive and inductive temporal link

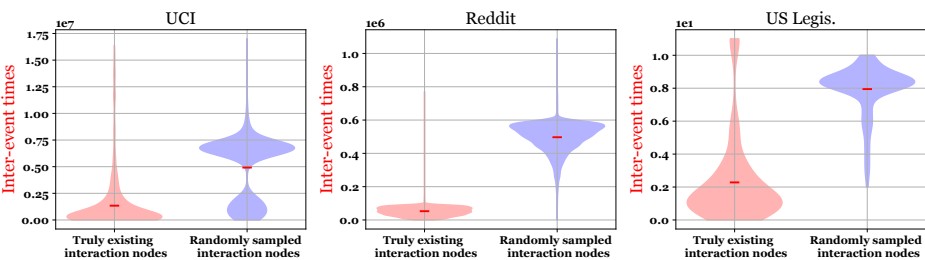

(a) Macro-level distribution of nodes' inter-event times across the entire timeline: Compared to the randomly sampled interaction nodes, most truly existing interaction nodes exhibit shorter periods of inactivity and short-term interaction bursts.

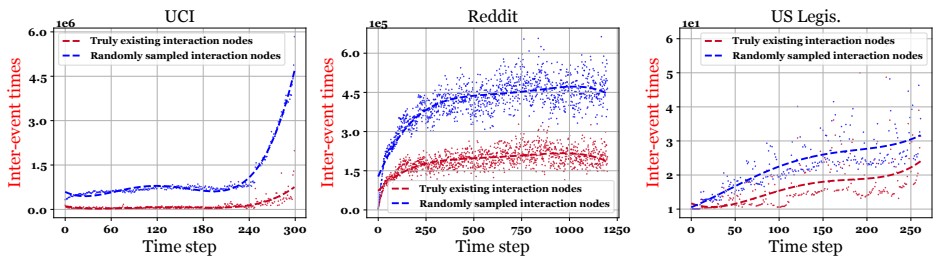

(b) Micro-level distribution of nodes' inter-event times over time steps: Short-term interaction bursts from truly existing interaction nodes consistently occur at most time steps.

Figure 9: Macro-level and micro-level empirical analyses for illustrating temporal clustering on the Wikipedia, UCI, and Contact datasets, respectively.

prediction tasks. Furthermore, most of baselines perform worse with the ROC-AUC metric than using the AP metric, suggesting that solely relying on the AP metric may not reflect the true prediction abilities of models. We emphasize that TG-Mixer demonstrates consistently strong performance in both AP and ROC-AUC metrics, affirming its effectiveness for temporal link prediction.

Table 9: ROC-AUC (%) results for temporal link prediction in the transductive setting. The best model performance is highlighted in %d and the second-best performance is denoted in %d .

| Models | Wikipedia | Reddit | LastFM | UCI | Flights | US Legis. | Contact |
|---|---|---|---|---|---|---|---|
| JODIE | 96.33 ± 0.07 | 98.31 ± 0.05 | 70.49 ± 1.66 | 90.44 ± 0.49 | 96.21 ± 1.42 | 82.85 ± 1.07 | 96.66 ± 0.89 |
| DyRep | 94.37 ± 0.09 | 98.17 ± 0.05 | 71.16 ± 1.89 | 68.77 ± 2.34 | 95.95 ± 0.62 | 82.28 ± 0.32 | 96.48 ± 0.14 |
| TGAT | 96.67 ± 0.07 | 98.47 ± 0.02 | 71.59 ± 0.18 | 78.53 ± 0.74 | 94.13 ± 0.17 | 75.84 ± 1.99 | 96.95 ± 0.08 |
| TGN | 98.37 ± 0.07 | 98.60 ± 0.06 | 78.47 ± 2.94 | 92.03 ± 1.13 | 98.22 ± 0.13 | 83.34 ± 0.43 | 97.54 ± 0.35 |
| CAWN | 98.54 ± 0.04 | 99.01 ± 0.01 | 85.92 ± 0.10 | 93.87 ± 0.08 | 98.45 ± 0.01 | 77.16 ± 0.39 | 89.99 ± 0.34 |
| TCL | 95.84 ± 0.18 | 97.42 ± 0.06 | 64.06 ± 1.16 | 87.82 ± 1.03 | 91.21 ± 0.06 | 76.27 ± 0.63 | 94.15 ± 0.09 |
| EdgeBank | 90.78 ± 0.00 | 95.37 ± 0.00 | 83.77 ± 1.16 | 77.30 ± 0.13 | 90.23 ± 0.13 | 62.57 ± 0.18 | 94.34 ± 0.00 |
| GraphMixer | 96.92 ± 0.03 | 97.17 ± 0.02 | 73.53 ± 0.12 | 91.81 ± 0.67 | 91.13 ± 0.06 | 76.96 ± 0.79 | 93.94 ± 0.02 |
| DyGFormer | 98.91 ± 0.02 | 99.15 ± 0.01 | 93.05 ± 0.10 | 94.49 ± 0.26 | 98.93 ± 0.01 | 77.90 ± 0.58 | 98.53 ± 0.01 |
| TG-Mixer | **99.83 ± 0.04** | **99.91 ± 0.01** | **96.72 ± 0.06** | **97.81 ± 0.26** | **99.59 ± 0.01** | **99.31 ± 0.36** | **99.64 ± 0.23** |

## C.7 RESULTS FOR MODEL CONVERGENCE AND GENERALIZATION CAPABILITIES

We provide the supplementary results for model convergence and generalization capabilities on other datasets in Figure 12. Our TG-Mixer still demonstrates exceptionally faster convergence and stronger generalization capabilities among various datasets compared to baselines.

Table 10: ROC-AUC (%) results for temporal link prediction in the inductive setting. The best model performance is highlighted in %d and the second-best performance is denoted in %d . Note that EdgeBank Poursafaei et al. (2022) predicts links between unseen nodes as negative, and therefore, it cannot be applied to the inductive setting.

| Models | Wikipedia | Reddit | LastFM | UCI | Flights | US Legis. | Contact |
|---|---|---|---|---|---|---|---|
| JODIE | $94.33 \pm 0.27$ | $96.52 \pm 0.13$ | $81.13 \pm 3.39$ | $78.80 \pm 0.94$ | $95.21 \pm 0.32$ | $58.12 \pm 2.35$ | $95.37 \pm 0.92$ |
| DyRep | $91.49 \pm 0.45$ | $96.05 \pm 0.12$ | $82.24 \pm 1.51$ | $58.08 \pm 1.81$ | $93.56 \pm 0.70$ | $61.07 \pm 0.56$ | $91.89 \pm 0.38$ |
| TGAT | $95.90 \pm 0.09$ | $96.98 \pm 0.04$ | $76.99 \pm 0.29$ | $77.64 \pm 0.38$ | $88.64 \pm 0.35$ | $48.27 \pm 3.50$ | $96.53 \pm 0.10$ |
| TGN | $97.72 \pm 0.03$ | $97.39 \pm 0.07$ | $82.61 \pm 3.15$ | $86.68 \pm 2.29$ | $95.92 \pm 0.43$ | $62.38 \pm 0.48$ | $94.84 \pm 0.75$ |
| CAWN | $98.03 \pm 0.04$ | $98.42 \pm 0.02$ | $87.82 \pm 0.12$ | $90.40 \pm 0.11$ | $96.86 \pm 0.02$ | $51.49 \pm 1.13$ | $89.07 \pm 0.34$ |
| TCL | $95.57 \pm 0.20$ | $93.80 \pm 0.07$ | $70.84 \pm 0.85$ | $84.49 \pm 1.82$ | $82.48 \pm 0.01$ | $50.43 \pm 1.48$ | $93.05 \pm 0.09$ |
| GraphMixer | $96.30 \pm 0.04$ | $94.97 \pm 0.05$ | $80.37 \pm 0.18$ | $89.30 \pm 0.57$ | $82.27 \pm 0.06$ | $47.20 \pm 0.89$ | $92.83 \pm 0.05$ |
| DyGFormer | $98.48 \pm 0.03$ | $98.71 \pm 0.01$ | $94.08 \pm 0.08$ | $92.63 \pm 0.13$ | $97.80 \pm 0.02$ | $53.21 \pm 3.04$ | $98.30 \pm 0.02$ |
| TG-Mixer | $\mathbf{99.82 \pm 0.05}$ | $\mathbf{99.88 \pm 0.01}$ | $\mathbf{95.97 \pm 0.59}$ | $\mathbf{96.81 \pm 0.51}$ | $\mathbf{99.09 \pm 0.01}$ | $\mathbf{98.97 \pm 0.89}$ | $\mathbf{99.68 \pm 0.27}$ |

## C.8 DETAILS FOR MODEL EFFICIENCY

Besides the discussions in Section 5.2, we also provide the averaged training wall-clock time of a single epoch under the optimal training hyper-parameters between the TG-Mixer and the baselines in Table 12. We observe that TG-Mixer requires significantly less time than most baselines and achieves the second lowest time consumption, demonstrating its high efficiency at each epoch. We emphasize that TG-Mixer achieves extremely better task performance than the fastest model (i.e., JODIE) as displayed in Table 1. We also notice that neither JODIE nor TG-Mixer consistently achieves the highest efficiency at each epoch across all datasets. This variability can be attributed to the fact that the dataset-dependent optimal hyper-parameters for these models may affect the training time consumption in practice. For example, a larger neighbor sample size $m$ necessitates longer training times. On the Wikipedia dataset, TG-Mixer achieves its best performance with a neighbor sample size of $m = 30$ while JODIE requires $m = 10$, leading to the high efficiency of JODIE. Instead, in the case of the Reddit dataset, TG-Mixer sets $m = 10$ and JODIE maintains the size ($m = 10$), where TG-Mixer achieves lower time consumption compared to that of JODIE. More optimal training hyper-parameter configure details of baselines can be found in Yu et al. (2023), and configuration details of TG-Mixer for training and evaluation are put in Section B.3 of the Appendix.

Table 11: Number of model parameters.

| Models | JODIE | DyRep | TGAT | TGN | CAWN | TCL | GraphMixer | DyGFormer | TG-Mixer |
|---|---|---|---|---|---|---|---|---|---|
| # Parameters ($\times 10^5$) | $\mathbf{1.96}$ | 6.92 | 10.53 | 9.64 | 160.91 | 8.84 | 6.42 | 10.87 | 5.23 |

Table 12: Wall-clock time (s) for training one epoch under the optimal training hyper-parameters.

| Models | Wikipedia | Reddit | LastFM | UCI | Flights | US Legis. | Contact | Avg. Rank |
|---|---|---|---|---|---|---|---|---|
| JODIE | $\mathbf{57.0}$ | 338.3 | $\mathbf{173.2}$ | $\mathbf{13.4}$ | 1040.6 | $\mathbf{16.0}$ | $\mathbf{232.5}$ | $\mathbf{1.50}$ |
| DyRep | 114.7 | 375.1 | 447.1 | 31.3 | 1696.3 | 18.6 | 488.9 | 3.75 |
| TGAT | 387.3 | 4241.3 | 3194.7 | 166.8 | 4897.8 | 156.0 | 6257.8 | 7.63 |
| TGN | 66.6 | 496.1 | 346.9 | 22.8 | 1249.3 | 38.3 | 520.2 | 3.63 |
| CAWN | 1082.0 | 3319.99 | 18235.5 | 613.5 | 18235.5 | 296.88 | 24729.2 | 8.87 |
| TCL | 66.6 | 377.7 | 476.5 | 25.1 | 751.3 | 36.3 | 906.8 | 4.13 |
| GraphMixer | 72.1 | 389.7 | 476.0 | 31.8 | 776.0 | 38.7 | 1119.9 | 5.25 |
| DyGFormer | 202.9 | 982.0 | 10406.2 | 94.9 | 12049.3 | 171.2 | 3885.2 | 7.50 |
| TG-Mixer | 60.6 | $\mathbf{271.2}$ | 450.2 | 21.9 | $\mathbf{725.7}$ | 27.4 | 725.6 | 2.63 |

## C.9 RESULTS FOR EXPRESSION ABILITY OF SILENCE DECAY MECHANISM

We visualize the complementary decay coefficients of our silence decay mechanism between positive links and negative links on the supplementary datasets in Figure 10. We can observe that our decay

mechanism provides a highly indiscriminative training signal, indicating the effectiveness of explicitly capturing temporal clustering for better temporal link prediction.

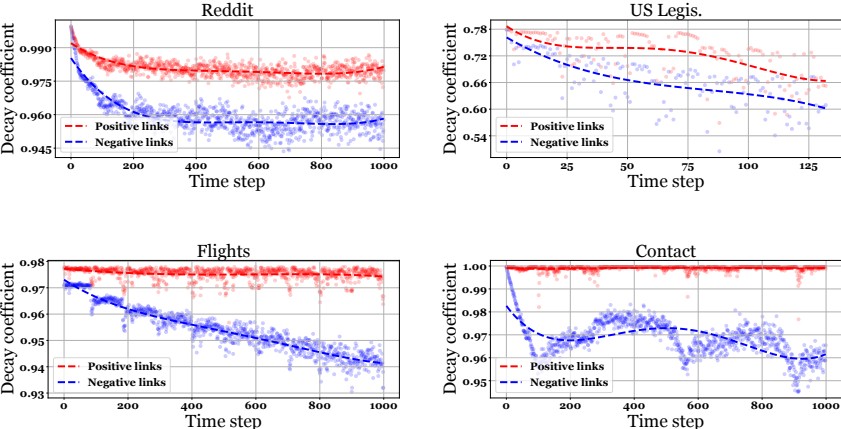

Figure 10: Comparisons of the decay coefficients produced by our silence decay mechanism between positive and negative links. Temporal clustering can offer a highly discriminative training signal.

## C.10   DETAILS FOR THE VERSATILITY ABILITY OF SILENCE DECAY MECHANISM

Our silence decay mechanism can also be easily adopted to boost existing sequential TGNs. Similar to TG-Mixer, these methods learn from 1-hop historical links and generate temporal representations by aggregating neighbor information with pooling operations, such as mean-pooling in GraphMixer Cong et al. (2023). In practice, we integrate the silence decay mechanism into these models by directly decaying the neighbor information before they perform the corresponding pooling operations. Supplementary results for inductive AP results are presented in Table 13.

From the results, we find that the silence decay mechanism that explicitly captures temporal clustering enables these models to achieve certain performance improvements, demonstrating its widespread applicability. Additionally, the most significant performance improvements are observed in some datasets with stronger temporal clustering, such as LastFM and Flights, validating the effectiveness and importance of temporal clustering.

Table 13: Inductive (%) AP results of existing sequential TGNs that are boosted by temporal clustering. Before "→" are the original results and after "→" are the boosting results via silence decay.

| Models | Wikipedia | Reddit | LastFM | Flights | US Legis. | Contact |
|---|---|---|---|---|---|---|
| TCL | 96.22 → **99.02** | 94.09 → **96.08** | 73.53 → **93.24** | 83.41 → **84.18** | 52.59 → **68.47** | 91.11 → **94.90** |
| GraphMixer | 96.65 → **98.13** | 95.26 → **97.00** | 82.11 → **95.48** | 83.03 → **85.38** | 50.71 → **66.49** | 90.59 → **96.51** |
| DyGFormer | 98.59 → **98.25** | 98.84 → **98.10** | 94.23 → **95.29** | 97.79 → **98.35** | 54.28 → **73.03** | 98.03 → **98.87** |
| TG-Mixer | **99.83** | **99.89** | **95.91** | **99.09** | **99.21** | **99.58** |

## C.11   RESULTS FOR EVALUATION OF THE NEIGHBOR SELECTION STRATEGY IN TG-MIXER

We provide the inductive AP results in Table 14 for validating different neighbor selection strategies with diverse neighbor sizes utilized in TG-Mixer. Employing the recent neighbor selection strategy enables TG-Mixer to achieve optimal performance, further indicating the crucial importance of preserving temporal clustering within raw data. Moreover, we find that the optimal sample size varies across different datasets, demonstrating that tuning the sample size is essential when performing temporal link prediction tasks.

Table 14: Inductive AP (%) results of diverse neighbor selection strategies in various sample sizes.

| Dataset | # Sample size | Recent sample | Random sample | Dataset | # Sample size | Recent sample | Random sample | Dataset | # Sample size | Recent sample | Random sample |
|---|---|---|---|---|---|---|---|---|---|---|---|
| Wikipedia | 10 | 99.30 | 94.00 | Reddit | 10 | **99.88** | **98.00** | UCI | 10 | 96.23 | 95.34 |
| | 20 | 99.49 | 94.31 | | 20 | 99.74 | 97.97 | | 20 | **96.81** | **95.35** |
| | 30 | **99.82** | **94.78** | | 30 | 99.27 | 97.63 | | 30 | 96.56 | 95.28 |
| | 50 | 99.74 | 94.62 | | 50 | 98.85 | 97.82 | | 50 | 96.10 | 95.29 |

## C.12 DETAILS FOR ABLATION STUDY

To better understand TG-Mixer, we conduct an ablation study to evaluate the effectiveness of TG-Mixer's components. Considering that the mainstream existing TGNs are mainly based on attention mechanisms, we first construct variants by replacing the information mixer introduced in Section 4 with the full/temporal attention mechanisms, where full attention is widely used in Transformers Vaswani et al. (2017) and temporal attention proposed by TGAT Xu et al. (2020) is widely employed in recent existing TGNs. We refer to these two variants as "**Full attention**" and "**Temporal attention**", respectively. Furthermore, note that MLP-Mixer Tolstikhin et al. (2021) serves as the backbone of TG-Mixer. Therefore, besides the two variants above, we let **MLP-Mixer** as one of our variants. Moreover, we integrate our silence decay mechanism and temporal mixer into MLP-Mixer, resulting in two variants: "**w/. silence decay mechanism**" and "**w/. temporal mixer**", respectively. Please notice that "w/. temporal mixer" is what we refer to as the TG-Mixer. Finally, to investigate the performance brought by the silence decay mechanism, we set the decay coefficient $g(\cdot) = 0$ of the silence decay mechanism in Equation 9, leading to the variant " TG-Mixer$_{g(\cdot)=0}$".

Supplementary inductive AP results for the ablation study are presented in Table 15.

Table 15: Inductive AP (%) results of ablation study.

| Techniques | Variants | Wikipedia | Reddit | UCI | Contact |
|---|---|---|---|---|---|
| Attention mechanism | Full attention Vaswani et al. (2017) | 96.87 | 96.46 | 77.31 | 90.85 |
| | Temporal attention Xu et al. (2020) | 96.22 | 97.09 | 79.54 | 95.87 |
| Information mixer | MLP-Mixer Cong et al. (2023) | 96.65 | 95.26 | 91.19 | 90.59 |
| | w/. silence decay mechanism | 98.13 | 97.00 | 97.08 | 96.51 |
| | w/. temporal mixer (*i.e., TG-Mixer*) | **99.83** | **99.89** | **96.56** | **99.58** |
| | TG-Mixer$_{g(\cdot)=0}$ | 97.14 | 95.93 | 93.18 | 93.26 |

## C.13 ADDITIONAL EXPERIMENTS UNDER DIFFERENT LEVELS OF TEMPORAL CLUSTERING

To further validate the robustness of TG-Mixer, we conduct a series of additional experiments focusing on scenarios characterized by different levels of temporal clustering, evaluating how TG-Mixer performs under disrupted or relatively low levels of temporal clustering. Specifically, we disrupt the interaction patterns by replacing the original interactions with randomly sampled noisy interactions at a certain perturbation rate Wang et al. (2021d), resulting in different levels of bursts among the interaction patterns. A higher perturbation rate corresponds to lower interaction bursts, indicating a lower level of temporal clustering among the temporal graphs. We employ GraphMixer Cong et al. (2023) and TCL Wang et al. (2021a) for comparisons due to their similar sequential model design, and the AP results for such an experimental setting are put in Figure 11.

From the results, we find that TG-Mixer can also capture basic temporal and structural information through the design of the information mixer introduced in Section 4, thus ensuring its performance even under disrupted or relatively low temporal clustering. Additionally, we also observe that TG-Mixer achieves similar results to GraphMixer at a high perturbation rate. This may be owing to the fact that both of these two models are based on MLP-Mixer architecture, where they have a similar ability to encode the basic temporal and structural information under low levels of temporal clustering. Furthermore, TCL suffers from more spurious information from high-order connections, thus leading to significant performance degradation at high perturbation rates.

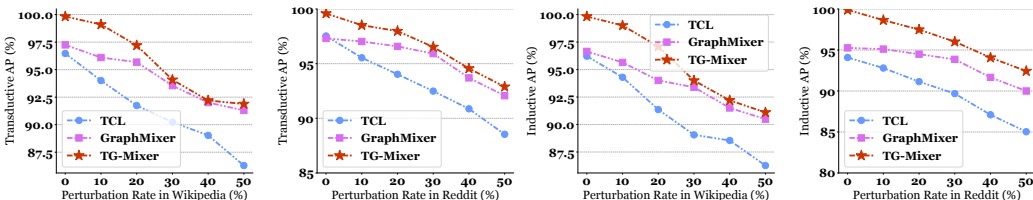

Figure 11: AP (%) results under different levels of temporal clustering. Higher perturbation rates indicate a lower level of temporal clustering. TG-Mixer can guarantee its performance even under disrupted or relatively low levels of temporal clustering.

# D  OTHER DISCUSSIONS

## D.1  RELATED WORK

**Temporal Graph Networks (TGNs).** According to the granularity of temporal information, dynamic graphs can be categorized into two primary types: Discrete-Time Dynamic Graphs (DTDGs), also known as Discrete Graphs, and Continuous-Time Dynamic Graphs (CTDGs), also referred to as Temporal Graphs Liang et al. (2023). Discrete graphs are characterized by multiple graph snapshots taken at fixed time intervals, where each snapshot represents a static graph, and all snapshots are ordered chronologically Xue et al. (2022); Kazemi et al. (2020). Existing methods for handling discrete graphs typically apply static graph techniques to each snapshot individually, with RNNs Pareja et al. (2020) or attention mechanisms Sankar et al. (2020); Cong et al. (2021) to capture temporal dependencies across different snapshots.

In contrast, temporal graphs Wen & Fang (2022); Tian et al. (2024b); Xiang et al. (2023) encapsulate more granular temporal information, with each edge explicitly annotated with interaction timestamps. This fine-grained temporal information allows the graph to evolve continuously over time. Consequently, temporal graphs present more challenges and opportunities. For capturing the dynamics and topologies within temporal graphs, Temporal Graph Networks (TGNs[5]) have been widely studied in recent years Goyal et al. (2020); Fan et al. (2021); Wang et al. (2021e); Rossi et al. (2020b); Gao & Ribeiro (2022); Chang et al. (2020); Pareja et al. (2020); Zhang et al. (2023); Chen et al. (2024); Zhang et al. (2024b). Most existing TGNs try to parameterize the time-dependent interaction patterns via their designed complex architectures, such as temporal graph convolutions Wang et al. (2021d;b); Alvarez-Rodriguez et al. (2021); Zhou et al. (2022); Luo & Li (2022); Zhang et al. (2024a), temporal random walks Wang et al. (2021c); Souza et al. (2022); Jin et al. (2022), and sequential models Wang et al. (2021a); Cong et al. (2023); Yu et al. (2023); Tian et al. (2024b). Although powerful, these methods overlook high-level temporal correlations and rhythm patterns among node interactions in raw data. In this paper, we introduce some empirical analyses to observe temporal clustering within node interaction rhythms and leverage this interesting phenomenon for advancing temporal link prediction. To this end, we propose a novel method to show the necessity of explicitly considering temporal clustering, which is achieved by two designs: a neighbor selection strategy and a temporal mixer with a silence decay mechanism.

**Existing Studies on Bursty Behavior.** We notice that some studies explore the effects of bursty interaction patterns in temporal networks Karsai et al. (2018); Hiraoka et al. (2020); Jo (2023). However, these works focus on designing statistical models to analyze the burst events and their impacts on the distribution characteristics Karsai et al. (2011), correlations Lambiotte et al. (2013); Min & Goh (2013), and propagation processes of inter-event times Barabasi (2005); Sheng et al. (2023). Instead, this paper leverages bursty interaction patterns to design a deep learning model that facilitates representation learning in temporal graphs. Additionally, these works are outside the ML community and primarily published in the field of physical sciences, which is definitely outside the scope of our paper.

**Difference Discussions.** To provide a clearer understanding of the concept of temporal clustering introduced in our paper, we briefly differentiate between "temporal clustering" and "clustering tasks among temporal graphs" Liu et al. (2023).

---

[5]For consistency, we follow Souza et al. (2022) in using "TGNs" to refer to a family of models for temporal graph representation learning. This term differs from the specific model "TGN" proposed by Rossi et al. (2020a).

Temporal clustering introduced in our paper reflects a distinctive *dynamics of interaction patterns* within temporal graphs: node interactions tend to occur in bursts, leading to a concentration or clustering of occurrence along the temporal dimensions. Different from temporal clustering, clustering tasks Yao & Joe-Wong (2021) are defined based on the *topological or connectivity tendencies of nodes* among graphs. In these tasks, nodes within a cluster are densely connected, while nodes across different clusters exhibit sparse connectivity. Recently, clustering tasks among temporal graphs Li et al. (2022); Liu et al. (2023; 2024) have emerged as a significant research trend in graph clustering. These studies often focus on adapting static clustering techniques to temporal graphs. For example, TGC Liu et al. (2023) investigates and introduces deep temporal graph clustering, which proposes a clustering method to suit the interaction sequence-based batch-processing of temporal graphs.

In this paper, we incorporate the high-level dynamics of interaction patterns (i.e., temporal clustering) for the temporal link prediction task, presenting a lightweight solution that achieves both effectiveness and efficiency. Our work is fundamentally different from the above clustering tasks in temporal graphs or other tasks that leverage topological connectivity density for performance improvements. Therefore, the above methods fall outside the scope of our paper, and we do not discuss their details.

### D.2 COMPLEXITY ANALYSIS

In addition to the experimental evaluation of model complexity in Section 5.2, we also provide a theoretical complexity analysis of the main components in TG-Mixer and selected baselines. Specifically, we choose three baselines: TGN Rossi et al. (2020a), CAWN Wang et al. (2021c), and DyGFormer Yu et al. (2023), as they represent the most representative method in temporal graph convolution models, temporal walk models, and sequential models, respectively.

Let $L$ represent the number of interaction sequences, $D$ denote the average node degree, $K$ indicate the number of encoding layers, and $m$ depict the sample size in the neighbor selection strategy. To simplify the calculations, we assume that the models' input, hidden, and output dimensions are uniformly set to $d$. TG-Mixer generates node representations by sampling the most recent historical links and then using a token mixer and a temporal mixer, leading to a computational complexity of $\mathcal{O}\left(mLd + mLd^2\right)$. TGN samples multi-hop most recent historical links, updates nodes' memory with RNNs, and employs a temporal attention mechanism to summarize link information, resulting in an overall complexity of $\mathcal{O}\left(Ld^2 + m^2KLd\right)$. CAWN encodes temporal walks with RNNs and aggregates these walks using an attention mechanism, which results in a computational complexity of $\mathcal{O}\left(nKLd^2 + n^2Ld\right)$ where $n$ represents the number of temporal walks. DyGFormer extracts all 1-hop historical links and employs a multi-layer Transformer encoder to aggregate link information, resulting in a complexity of $\mathcal{O}\left(KLD^2d + KLd\right)$.

### D.3 LIMITATIONS

One potential limitation of TG-Mixer is the cold start issue for learning temporal clustering at the beginning of the training (discussed in Section 5.2). However, after several epochs, TG-Mixer effectively captures and leverages the powerful predictive signals derived from explicitly considering temporal clustering, and achieves outstanding performance compared to the baselines.

Another potential limitation of TG-Mixer is its neighbor selection strategy that samples from 1-hop historical links. It may be suboptimal when handling certain scenarios in which nodes' high-order relationships are crucial. However, simply incorporating multi-hop links into TG-Mixer would significantly increase computational costs and weaken its ability to capture temporal clustering. There is significant potential in designing both efficient and effective methods to capture nodes' high-order relationships for better temporal link prediction.

### D.4 DISCUSSIONS OF MODEL PERFORMANCE ON THE US LEGIS. DATASET

The temporal link prediction results show that TG-Mixer outperforms baselines on the US Legis. dataset with a large margin. In this section, we briefly discuss the potential reasons for it. This superior performance is due to the composite effect of at least two key factors:

**Smaller timestamp gap.** US Legis. has a significantly smaller timestamp gap (the difference between the maximum and minimum timestamps) compared to other datasets. For example, US

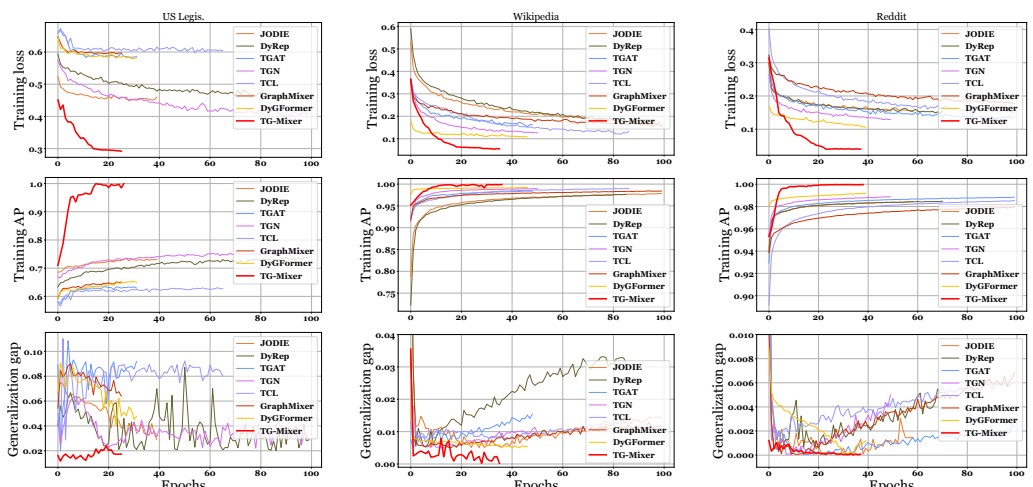

Figure 12: Comparisons of the training loss, training AP, and the generalization gap over epochs when training models. TG-Mixer demonstrates faster convergence and stronger generalization capabilities.

Table 16: Characteristics of timestamp gap and interaction density across datasets used in this paper.

|  | Wikipedia | Reddit | LastFM | UCI | Flights | US Legis. | Contact |
|---|---|---|---|---|---|---|---|
| **Timestamp Gap** | $6.78 \times 10^5$ | $2.68 \times 10^6$ | $1.37 \times 10^8$ | $1.67 \times 10^7$ | 120 | 11 | $2.41 \times 10^6$ |
| **Interaction Density** | $1.12 \times 10^{-4}$ | $9.15 \times 10^{-5}$ | $5.09 \times 10^{-4}$ | $5.35 \times 10^{-5}$ | 1.20 | 19.74 | 0.43 |

Legis. has a timestamp gap of 11 while the dataset with the second smallest timestamp gap, Flights, has a timestamp gap of 120. Other datasets exhibit larger timestamp gaps. Baseline models process interactions separately and rely solely on a time-encoding function to encode temporal information. This function could fail to provide distinguishable encoding under a smaller timestamp gap, leading to the potential over-smoothing issue Yu et al. (2023) and achieving suboptimal performance in the US Legis. dataset. Instead, TG-Mixer captures high-level temporal correlations among interactions by considering temporal clustering information, making it less susceptible to performance degradation caused by a small timestamp gap.

**Higher interaction density in temporal dimensions.** Node interaction density in temporal dimensions refers to the number of node interactions at each time step, which is computed by #Node Degree / #Time Steps. US Legis. exhibits a significantly higher density compared to other datasets. For example, US Legis. has a density of 19.74 while the second highest, Flights, has a density of 1.20. Other datasets show even lower densities. A higher interaction density indicates that nodes have a large number of interactions occurring simultaneously, resulting in prominent interaction bursts and stronger temporal clustering. Consequently, baseline models do not benefit from such temporal clustering whereas TG-Mixer fully capitalizes on it, thus achieving exceptional performance.

The corresponding characteristics of timestamp gap and interaction density in the temporal dimension are summarized in Table 16.

