# OpenReview forum: "Interactions Exhibit Clustering Rhythm: A Prevalent Observation for Advancing Temporal Link Prediction"
_ICLR.cc/2025/Conference — Submitted to ICLR 2025_

### Official Review · Reviewer_TU2Y · 2024-10-31

**Soundness:** 2
**Presentation:** 1
**Contribution:** 2
**Rating:** 5
**Confidence:** 3

**Summary:**

In this paper, the authors considered the temporal link prediction task on continuous-time dynamic graphs (CTDGs). They first demonstrated that temporal clustering patterns (also defined as node interaction rhythms in this paper) are obvious latent features of various dynamic graphs, based on a series of empirical analysis on some real datasets. A TG-Mixer method was then proposed to fully explore these patterns with some designs of most recent historical links sampling, silence decay mechanism, etc.

**Strengths:**

**S1**. The authors conducted a great amount of experiments to demonstrate the motivations and validate the effectiveness of the proposed method.

**S2**. The proposed method can achieve good experiment results.

**Weaknesses:**

**W1**. The overall presentation of this paper is poor. There are many unclear, confusing, or even inconsistent statements, which make the motivation of this paper weak.

  In this paper, use 'temporal graph networks' (TGNs) to represent deep learning models designed to handle dynamic graphs. However, 'graph' and 'network' are usually synonyms in some literature. 'Temporal graph networks' may cause some ambiguity issues. As I can check, TGN is also the name of an existing method, while the authors used TGNs to represent a set of methods.

  At the very beginning of this paper, the authors claimed that existing methods 'process interactions separately without considering the inherent high-level temporal correlations among interactions between nodes' and thus 'fail to effectively capture the invaluable patterns of interaction dynamics'. Such a statement about the research gap is quite weak. It is unclear for me what do 'high-level temporal correlations among interactions' and 'interaction dynamics' refer to. There are no any toy examples to demonstrate these concepts. From my perspective, directly reaching such a conclusion without giving further interpretations or toy examples (e.g., due to what mechanism existing methods fail to capture such dynamics) is also insufficient, which makes the the motivation of this paper very weak.

  The authors repeatedly use 'temporal clustering' and 'node interaction rhythms', which are high-level concepts seldom used in related literature. Even though they gave the corresponding definitions in Section 2, they are still hard to understand. For some new but important concepts or terminologies, it is recommended to give some toy examples (e.g., a dynamic graphs with only a few nodes) in addition to their formal definitions, especially at the very beginning of a paper.

  The authors also claimed that Fig. 2 (a) confirms the presence of temporal clustering patterns but why? Since there are no interpretations or examples to demonstrate how to measure the temporal clustering patterns, it is also insufficient to reach such a conclusion, from my perspective. It is also similar for the discussions regarding Fig. 2 (b).

  For the further discussions regarding the empirical analysis w.r.t. Fig. 2 (a) and (b), the authors claimed that 'recent historical links, carry the latest rhythm pattern information, are crucial for prediction future links'. From my perspective, we can only reach a conclusion that 'recent historical links' are obvious latent futures in dynamic graphs based on Fig. 2 (a) and (b), but cannot say 'they are crucial for prediction future links', since Fig. 2 (a) and (b) are not directly related to the quality of temporal link prediction.

***

**W2**. Some details of the proposed methods are unclear or with weak motivations.

  From my perspective, the connection between the empirical analysis in Section 3 and some designs in Section 4 is weak. For instance, it is unclear for me why sampling the most recent historical links can preserve interaction rhythm patterns and why conventional sampling strategies may fail to preserve these patterns.

  There may exist some flaws of only sampling top $m$ most recent historical. It is well-known that some real dynamic graphs may follow typical power-law distributions. The power-law distribution of node degrees may indicate that there are some nodes with many historical links and some other nodes with very few historical links. $m$ is a constant for all the nodes in the proposed method. This design may result in an exception that there are some nodes without any sampled historical links (i.e., with degree less than $m$).

  In Eq. (3), it seems that ${\bf{C}}_{decay}^{t}$ is a vector associated with a node $u$, but there is no subscript or superscript of $u$ for this vector.

***

**W3**. The theoretical analysis in Appendix A is weak or even needless.

  From my perspective, the connection between Eq. (8) and Eq. (9) is not exactly as the authors claimed in Appendix A. In Eq. (9), the first term ${\bf{C}}_{rhythm}^t$ is used by all the nodes, as described in Section 4.

Whereas, in Eq. (8), the first term is associated with a node $u$. The connection between the summation in the second term of Eq. (8) and $g(s_u^t) {\bf{C}}_{temp}^t$ in Eq. (9) is also not clearly interpreted.

  The discussions in Appendix A cannot be considered as a theoretical contribution, since it just simply listing two existing known definitions (i.e., Definition 1 and Definition 2). There are no original theoretical results and corresponding proofs.

***

**W4**. Some experiments settings and results are unclear, which need further clarification.

  The the definition of the X-axis in Fig. 2 (a) is not given. As a result, it is unclear for me what does 'macro-level distribution' mean and how to correctly read Fig. 2 (a). The unit of the y-axis (i.e, silence duration) in Fig. 2 (a) and (b) is not given.

**Questions:**

See **W1**-**W4**.

---

> ### Author Response · Authors · 2024-11-20
> **Response to Reviewer TU2Y [Part 1]**
>
> Many thanks to Reviewer TU2Y for the thorough comments. We have revised our paper accordingly.
>
> > **W1-1:** Explanation of "TGNs".
>
> *Quick Answer: We would like to clarify that "TGNs" is widely used in the community, and we follow such terminology for readability.*
>
> * **Consistency with the research community.** "TGNs" is a commonly used terminology to refer to a family of models designed for learning node representations in temporal graphs [1, 2, 3].
>
> * **Differences between "Graph" and "Network."** "Graph" generally refers to the structured data with nodes and edges, while "Network" typically describes the neural network in model structures such as KAN (Kolmogorov-Arnold Networks) [4].
>
> [1] Towards Adaptive Neighborhood for Advancing Temporal Interaction Graph Modeling, KDD, 2024.
> [2] Provably Expressive Temporal Graph Networks, NeurIPS, 2022.
> [3] Adaptive Data Augmentation on Temporal Graphs, NeurIPS, 2021.
> [4] Kan: Kolmogorov-arnold Networks, Arxiv, 2024.
>
> > **W1-2:** Illustration of the motivation for our work.
>
> *Revision: As revised in Lines 49-68, we have clarified the motivation for our work.*
>
> We provide the overall explanation of our motivation here.
>
> * **Limitations of existing works.** As noted in Lines 49-68, existing works often focus on designing different model architectures to parameterize the interaction patterns in temporal graphs. Although powerful, these frameworks rely heavily on intricate computational processes to approximate their parameterized interaction patterns. Such a procedure tends to overlook the potentially heuristic, realistic patterns inherent in the raw data, leading models to depend increasingly on more complex architectures to fit fragmented details for performance improvement.
> * **Research gap between our work and existing works.** Unlike the mainstream idea of designing complex model architectures, we introduce statistical empirical analyses and present the first attempt to investigate the heuristic temporal correlation among interactions. Building on these observations, we propose the first model that explicitly captures burstiness information for temporal link prediction.
>
> > **W1-3:** Illustrative examples for temporal clustering.
>
> *Revision: As **revised in Line 66 of Figure 1**, we have included an illustrative example that vividly demonstrates temporal clustering patterns.*
>
> Thanks for your suggestion! In Figure 1, we provide a toy temporal graph and plot the interaction rhythms for three example nodes. This example demonstrates how the behavior of a specific node in temporal graphs exhibits burstiness, providing a clearer and more intuitive understanding of temporal clustering.
>
> > **W1-4:** Explanation of the measurement and conclusion in empirical analyses.
>
> *Quick Answer: We would like to clarify that we have already introduced the measurement for temporal clustering and performed a reliable analysis to derive our conclusions in empirical analyses.*
>
> * **Measurement for temporal clustering.** As stated in Line 187, to quantify temporal clustering, we compute the average inter-event times for interaction nodes (notably, we have replaced the term "silence duration" with "inter-event times" to improve readability). Besides, in Line 190, we explain the motivation behind this method: "Interaction nodes exhibiting strong temporal clustering tend to experience shorter inter-event times."
>
> * **Analysis for conclusions.** As mentioned in Lines 200-208 and 236-251, we have provided a thorough analysis to draw the conclusions. For example, as discussed in Line 201, most truly existing interaction nodes (the red distribution in Figure 3(a)) exhibit heavy-tailed distribution with shorter inter-event times. This indicates that nodes in temporal graphs reveal burstiness in their behaviors with shorter periods of inactivity, which strongly supports the presence of temporal clustering.
>
> > **W1-5:** Relevance between empirical analyses and link prediction.
>
> *Quick Answer: Kindly note that the comparison between truly existing and randomly sampled interaction nodes **directly aligns with the task of distinguishing positive and negative links in link prediction**.*
>
> * **Relevance to link prediction.** As outlined in Lines 154-159, we perform empirical analyses by comparing the differences between truly existing and randomly constructed interactions. This setup directly corresponds to the link prediction task, which aims to distinguish positive links (those that truly exist) and negative links (those that do not).
> * **Insights for link prediction.** As revised in Lines 243-245, we summarize the key factors of temporal clustering for temporal link prediction and explain them as (i) if a given node has recent interactions, then it is more likely to interact at the current timestamp; and (ii) if the node has only distant past interactions or even no historical interactions, it is less likely to interact now.

---

> ### Author Response · Authors · 2024-11-20
> **Response to Reviewer TU2Y [Part 2]**
>
> > **W2-1:** Explanation of the motivation behind neighbor selection strategy.
>
> *Quick Answer: We would like to clarify that we do **not** claim that existing works fail to preserve interaction rhythm patterns. Rather, their limitation lies in "incorporating spurious or noisy information", which prevents them from **effectively preserving the latest and most relevant** interaction patterns.*
>
> In Lines 264-268, we have acknowledged that the existing neighbor selection strategy can help preserve interaction rhythm patterns. The limitation of these techniques is that "they may risk incorporating spurious or noisy information either from high-order connections or long-outdated histories". To this end, we sample the most recent historical links, preserving the latest, most relevant interaction patterns for accurate future link prediction.
>
> > **W2-2:** Explanation of the sample size $m$.
>
> We kindly draw your attention to **Line 280**, for nodes with fewer than $m$ historical links, we select all available links and pad the corresponding matrix to $m$ dimensions. This technique is also adopted in existing studies [1, 2].
>
> [1] Towards Adaptive Neighborhood for Advancing Temporal Interaction Graph Modeling, KDD, 2024.
> [2] Towards Better Dynamic Graph Learning: New Architecture and Unified Library, NeurIPS, 2023.
>
> > **W2-3:** Explanation of $\mathbf{C}_{\text{rhythm}}^t$.
>
> Kindly note that, as mentioned in **Line 305**, the rhythm vector $\mathbf{C}_{\text{rhythm}}^t$ is shared communally across all nodes and will be updated globally and chronologically. Therefore, to avoid confusion, we do not use the superscript $u$.
>
> > **W3:** Explanation of theoretical analysis.
>
> *Quick Answer: We kindly believe that the theoretical analysis provides a strong foundation for supporting the proposed silence decay mechanism.*
>
> In the theoretical analysis presented in Section A, we define the Hawkes process and temporal clustering effects, which form the basis for deriving Eq. 8. Our silence decay mechanism, introduced in Eq. 9, is closely aligned with the framework outlined in Eq. 8.
>
> As shown in Eq. 9, when generating the representation of node $u$ at timestamp $t$, we retrieve $\mathbf{C}_ {\text{rhythm}}^t$ and use it as the underlying rate of interaction occurrences ($\mu_u^t$ in Eq. 8). The decay function $\text{g}(\cdot)$ acts as the kernel function to capture the time decay effects ($\rho(\cdot)$ in Eq. 8), and the $\mathbf{C}_ {\text{temp}}^t$ represents the excitation amount for future behavior of node $u$ ($\gamma_u^{t'}$ in Eq. 8). For further details, please refer to Lines 756-801.
>
> > **W4:** Clarification of Figure 3 (Figure 2 in the original paper).
>
> **Definition of X-axis.** We would like to clarify that as noted **in Lines 160-188**, we have provided the formulated definition of truly existing and randomly sampled interaction nodes in our original paper. Specifically, truly existing interaction nodes refer to node pairs that exist in temporal graphs, while randomly sampled interaction nodes are obtained by substituting the original interaction nodes with randomly sampled nodes. For further details, please refer to Lines 160–188.
>
> **Meaning of "macro-level".** Kindly note that as mentioned **in Lines 250-257**, we have already explained the meaning of "macro-level" in our original paper. To be specific, we claim that macro-level analyses capture node interaction rhythms across the entire timeline, rather than focusing on micro-level time steps. This is why we also conduct micro-level analyses for more granular insights.
>
> **Units of Y-axis.** Regarding the units of inter-event times, some datasets, such as Wikipedia, normalize timestamps when they are made publicly available. As a result, the timestamps in these datasets do not have explicit units. Furthermore, as shown in Table 6, the time granularities vary across different datasets. Therefore, following the analyses in [1], we focus solely on the numerical values of the timestamps without considering units.
>
> Thank you once again for your invaluable comments. Your efforts have been instrumental in helping us improve our work significantly.
>
> [1] Towards Better Evaluation for Dynamic Link Prediction, NeurIPS, 2022.

---

> > ### Comment · Reviewer_TU2Y · 2024-11-27
> >
> > I appreciate the authors' responses and revisions, which to some extent address some of my concerns. However, some of my concerns remain.
> >
> > For W1, although TGNs was used in 3 papers as mentioned by the authors, I still do not agree with using this terminology, due to the potential ambiguity issue among 'graph', 'network' (may refer to a graph or deep neural network model), 'TGNs' (a set of methods), and 'TGN' (a baseline in experiments).
> >
> > Although ${\bf{C}}_{\rm{rhythm}}^{t}$ is a vector is shared by all nodes,
> >
> > my concern is that ${\bf{C}}_{\rm{decay}}^{t}$ should be associated with a node $u$
> >
> > (even though it is derived based on ${\bf{C}}_{\rm{rhythm}}^{t}$,
> >
> > of course it must since ${\bf{C}}_{\rm{rhythm}}^{t}$ is shared by all nodes).
> >
> > In this sense, ${\bf{C}}_{\rm{decay}}^{t}$ should have a subscript or superscript regarding $u$.
> >
> > For W2, I still reserve my negative opinion regarding the theoretical analysis in Appendix A. From my perspective, the simple multiplication of output of two neural networks (i.e., $g(s_u^t) {\bf{C}}_{\rm{temp}^t}$)
> >
> > is not equivalent to the summation $\sum\limits_{t' < t} {\gamma _u^{t'}\rho (t - t')}$ in Eq. (8).
> >
> > Appendix A cannot be considered as a rigorous theoretical analysis, since it just simply listing two existing known definitions and simply claimed that two equations are with similar (but not exactly equivalent) format.
> >
> > After comprehensively considering the authors responses, I decide to raise my score from 3 to 5.

---

> > > ### Author Response · Authors · 2024-11-27
> > > **Thanks for raising the score!**
> > >
> > > Thank you for raising your overall score and for recognizing **the proposed method and its effectiveness**, as well as for acknowledging other aspects of our rebuttal and revisions, particularly in **motivation, illustrative examples, and the explanation of empirical analyses**.
> > >
> > > To address your remaining concerns, we provide more detailed responses as follows.
> > >
> > >
> > >
> > > > **W1:** Explanation of terminology and revision to formulation.
> > >
> > > Thank you for your comments and suggestions. We have made the following revisions to ensure that the expressions in our paper are more precise.
> > >
> > > * **Clarification of the term 'TGNs' in Lines 1348-1349.** As stated in W1-1 of our rebuttal, we adopt the widely used term "TGNs" in our paper for consistency and readability. To avoid other potential misunderstandings, as revised in Lines 1348–1349, we explicitly highlight the difference between the term 'TGNs' and the concrete 'TGN' model, ensuring a strict differentiation throughout our paper.
> > > * **Revising formulation from the $\mathbf{C}_ {\text{decay}}^t$ to $\mathbf{C}_ {u, \text{decay}}^t$ in Lines 318 (Eq. 4) and 344 (Eq. 5).** We apologize for previously misunderstanding your concern. We fully agree with your point that while $\mathbf{C}_ {\text{rhythm}}^t$ is shared among all nodes, $\mathbf{C}_ {\text{decay}}^t$ is associated with a node $u$. Therefore, we add the subscript $u$, correcting $\mathbf{C}_ {\text{decay}}^t$ into $\mathbf{C}_ {u, \text{decay}}^t$.
> > >
> > >
> > > > **W2:** Explanation of theoretical analysis in Appendix A.
> > >
> > > Thanks for your invaluable comments, and we provide a further explanation of our theoretical analysis.
> > >
> > > * **Motivation: Theoretical analysis underscores that the computational process of the silence decay mechanism adheres to the fundamental principles of the Hawkes Process.** As mentioned in Line 797, our theoretical analysis emphasizes that temporal clustering information is captured in a self-excited manner, which aligns with the core theories of the Hawkes Process. Therefore, we provide a formulaic comparison between the silence decay mechanism and the foundational theory of the Hawkes Process, offering a compelling validation of its theoretical soundness.
> > > * **Remark: Our silence decay mechanism focuses on the effects of the Hawkes Process from the last historical interaction to capture temporal clustering information.** As mentioned in Line 791, the summation $\sum_ {t' < t}\gamma_{u}^{t'} \rho(t - t')$ in Hawkes Process captures the excitation from **all historical interaction** before $t$. On the other hand, the excitation in our silence decay mechanism $\operatorname{g}\left(s_u^t\right) \mathbf{C}_{\text{temp}}^t$ contains the excitation information of temporal clustering from **the last interaction** before $t$. We believe they are equivalent because temporal clustering information is embedded in inter-event times, with each interaction focusing solely on its preceding interaction. Therefore, our silence decay mechanism could capture temporal clustering through the Hawkes Process from the last historical interaction.
> > >
> > > We are deeply grateful for your meticulous comments, which have significantly improved our paper. Your comprehensive review has been incredibly insightful, and we have learned a great deal from it. Thank you once again for raising our score!

---

### Official Review · Reviewer_AAW9 · 2024-10-31

**Soundness:** 3
**Presentation:** 3
**Contribution:** 2
**Rating:** 5
**Confidence:** 4

**Summary:**

This study empirically shows that nodes often interact with others in bursts. Building on this observation, the authors introduce TG-Mixer, a novel model that captures these temporal clustering patterns. While its framework is similar to GraphMixer, TG-Mixer introduces a Temporal Mixer module that penalizes nodes for prolonged inactivity. TG-Mixer demonstrates superior performance and higher efficiency compared to other baseline models across several datasets.

**Strengths:**

1. This work aims to uncover the temporal patterns in temporal graphs, which is a novel perspective that few relative studies address.
2. TG-Mixer demonstrates outstanding performance across several datasets, including US Legis and LastFM, and is more efficient than baseline models.
3. The writing is clear and easy to follow.

**Weaknesses:**

1. The key insight of this work is that the popular nodes are likely to interact with others or be interacted with. However, in many real-world applications of temporal graphs, such as recommender systems, popularity is already a significant factor. Thus, the more challenging problem is to rank these popular items. Simply incorporating popularity offers limited contributions to temporal graph learning, so it would strengthen the study to further demonstrate the power of TG-Mixer to distinguish popular nodes. For example, these empirical comparisons could be conducted:
     1. comparing TG-Mixer and the heuristic method that always chooses the most popular destination nodes.
     2. using the MRR metric with the 100 most **popular** destination nodes as negative destination nodes.
     3. using historical neighbors as negative destination nodes, as proposed in [R1].
2. The authors have not provided the complete implementation code in the supplementary materials, notably missing the `TimeMixer` component, which is central to the proposed method. I recommend that the authors make the full code for this work available.

[R1] Towards better evaluation for dynamic link prediction.

**Questions:**

1. In the experiments, do the negative edges share the same source nodes as the positive edges? If not, it would improve consistency to fix the source nodes for both positive edges and their corresponding negative edges across all experiments.
2. Among the TGB datasets, tgbl-review and tgbl-comment are considered challenging according to existing empirical studies. How does TG-Mixer perform on these two datasets?

---

> ### Author Response · Authors · 2024-11-20
> **Response to Reviewer AAW9**
>
> Many thanks to Reviewer AAW9 for providing an insightful review.
> > W1-1: Explanation of temporal clustering.
>
> *Quick Answer: Kindly note that **temporal clustering studies the burstiness of interactions for specific nodes** rather than node popularity.*
>
> **Difference between burstiness and popularity (node degree).** Bursty interactions within short periods emphasize relative interaction frequencies compared to other periods and are not directly correlated with node degree. For example, as mentioned in Line 77, a user's purchasing behavior might occur around weekends while exhibiting few activities during weekdays, even if the total number of purchases of this user is low.
>
> **Insights of temporal clustering for link prediction.** As revised in Lines 242-245, temporal clustering encodes the following insights for link prediction:
> * (i) If a given node has recent interactions, then it is more likely to interact at the current timestamp;
> * (ii) If the node has only distant past interactions or even no historical interactions, it is less likely to interact now.
>
> > W1-2: Evaluation under different negative sampling strategies.
>
> *Quick Answer: TG-Mixer **performs best across four negative sampling strategies**, reinforcing its effectiveness.*
>
> *Revision: We have incorporated these discussions in Lines 972-1007.*
>
> **Empirical Analysis.** In Line 908, we evaluate model performance using random sampling strategy. We note that recent work [1] has introduced advanced evaluation methods for temporal link prediction. To fully assess the models, we adopt a more challenging evaluation with four negative sampling strategies and rank-based metrics. Descriptions of these strategies are put in the "Implementation Details".
>
> **Experimental Evaluation.** Due to char limitations, we provide key findings and results here. Other analyses can be found in Lines 972-1007.
>
> * TG-Mixer **demonstrates strong performance** across four negative sampling strategies, further proving its robustness.
> * Inductive negative sampling often leads to the most performance drop. This is likely because this strategy samples unseen nodes to construct negative links, making it more challenging for models to distinguish temporal links.
>
> Table 1: A subset of transductive MRR results on Reddit.
> ||JODIE|TGAT|TGN|DyGFormer|TG-Mixer|
> |:-:|:-:|:-:|:-:|:-:|:-:|
> |random|70.24|64.24|72.95|81.47|**88.10**|
> |historical|61.83|60.77|62.81|70.16| **73.80**|
> |inductive|63.18|55.18|57.94|68.18| **70.51**|
> |degree-aware|68.94|59.17|60.19|71.92|**74.24**|
>
> **Implementation Details.** In addition to random negative sampling, we follow [1] and use **historical negative sampling** (sampling negative links that have been observed) and **inductive negative sampling** (sampling negative links that are unseen during training). To mitigate potential biases from popular nodes, we conduct **degree-aware negative sampling** by sampling negative nodes with the probability of $d/N$, where $d$ is the node's current degree and $N$ is # historical interactions.
>
> [1] Towards Better Evaluation for Dynamic Link Prediction, NeurIPS, 2022.
> > W2: Supplementary material.
>
> Kindly note that we have updated the complete codes for TG-Mixer. Please check it.
> > Q1: Explanation of negative sampling.
>
> As revised in Lines 908-911, we follow the same configurations as DyGLib, where we fix the source nodes and sample an equal number of destination nodes to construct negative links. Thus, our negative and positive edges share the same source nodes.
> > Q2: Evaluation on more datasets in TGB.
>
> *Quick Answer: TG-Mixer **exhibits superior performance across the additional datasets** in TGB.*
>
> **Empirical Analysis.** In the original paper, we conduct experiments on the two selected datasets because many baselines may not be implemented on other datasets due to computational issues ('NA' in the following Table). To further validate the model performance, as revised in Line 956, we add three datasets: Review, Comment, and Flight.
>
> **Experimental Evaluation.** We provide a subset of experimental results here. For the complete results, please refer to Table 7 of Line 956.
> * TG-Mixer **consistently demonstrates superior performance** on the additional datasets, highlighting its robustness and strong capabilities.
> * The most significant performance improvements are observed in the Review and Flight datasets. This can be attributed to their higher interaction density (computed as #node degree / #time steps, as described in Line 1408), where nodes tend to feature a larger number of simultaneous interactions. Such patterns result in pronounced burstiness and stronger temporal clustering, thus enhancing the effectiveness of TG-Mixer.
>
> We hope our revised paper has addressed your concerns.
>
> Table 2: A subset of MRR results on TGB.
> ||DyRep|TGN|GraphMixer|DyGFormer|TG-Mixer|
> |:-:|:-:|:-:|:-:|:-:|:-:|
> |Review|40.06|37.48|36.89|22.49|**49.92**|
> |Comment|NA|NA|76.17|67.03|**82.17**|
> |Flight|NA|NA|77.66|NA|**84.88**|

---

> > ### Comment · Reviewer_AAW9 · 2024-11-23
> >
> > Thank you for the explanations and the detailed additional experiments. While I am satisfied with most experiments, I still have concerns regarding the proposed weakness 1, which I will elaborate on below.
> >
> > I understand your motivation that "if a given node has recent interactions, then it is more likely to interact at the current timestamp". However, this motivation may not significantly aid the link prediction task in real-world scenarios. For instance, if we consider recommending items to users, the primary challenge lies in predicting the most relevant item among the recently popular items for a specific user.
> > It is important to note that the concept of “popularity” is inherently tied to timing—only items with many recent interactions are considered popular.
> > While it is evident that items with many recent interactions are more likely to be the desired item for a user, identifying the specific item for that user remains challenging. This is particularly relevant in real-world scenarios.
> > For example, on an online shopping platform, there may be numerous popular items that many users have recently purchased. However, recommending the most popular computer to a user who has never purchased electronics and primarily buys clothing would be ineffective.

---

> ### Author Response · Authors · 2024-11-23
> **Further Explanations for W1**
>
> We sincerely appreciate your recognition of our **model novelty, conducted experiments, and writing presentation**, as well as your acknowledgment of the other aspects of our rebuttal.
>
> Based on your expanded explanation for "popularity", we provide a more detailed response to W1 under the view of the Recommender System:
>
>
> * **Temporal clustering provides inductive bias or auxiliary shortcuts for capturing preferences of users, rather than simply recommending popular items.** We understand your concern, which can be explained as: recommending a suitable item for a user should prioritize the relevant items that reflect its preference (e.g., clothing in your provided example), rather than only recent popular nodes (e.g., computers in your example). However, TG-Mixer has already considered both, as it constructs neighborhoods to summarize node preference (achieved by the token mixer in Line 292) and encodes temporal clustering information to amplify and mix the neighborhood information for representations (achieved by the information mixer in Line 303). Therefore, TG-Mixer may tend to recommend relevant and recent popular nodes to the user, thereby enhancing recommendation performance.
> * **Predicting new links may indeed pose challenges for TG-Mixer, which is a recognized potential limitation in Line 945.** As discussed in Line 945, new links (e.g., a user who has never purchased electronics in your example) may conflict with the motivation behind temporal clustering, as the node pairs of these links lack recent historical interactions but still represent positive links. Consequently, in datasets with a significant proportion of new links (e.g., Coin in Table 7 of Line 956), TG-Mixer achieves performance that is competitive with the baselines, which could be a limitation of TG-Mixer. However, as highlighted in Line 951, we emphasize that TG-Mixer performs best in the majority of datasets (eleven in total), particularly those where simultaneous interactions are more prevalent.
>
>
> Once again, thank you for your valuable insights and the recognition of our work. If you have any further questions or concerns, please do not hesitate to reach out; we would be more than happy to address them. **:)**

---

> > ### Comment · Reviewer_AAW9 · 2024-11-25
> >
> > Thank you for your further explanation. My previous concern has been addressed. However, I noticed that the implementation of your proposed method is still incomplete in the revised supplementary materials. I encourage you to upload the complete implementation, including detailed commands, to ensure the results in your paper can be fully reproduced. If I am able to verify the reproducibility of your results, I will consider raising my score.

---

> > > ### Author Response · Authors · 2024-11-26
> > > **Response to Reviewer AAW9 for W2**
> > >
> > > We sincerely appreciate your valuable support during both the review and discussion phases, and we are deeply grateful for your recognition of our work, particularly regarding its **innovation, model design, and comprehensive experiments.**
> > >
> > > We are thrilled that you recognize our efforts and consider raising the score!
> > >
> > > **Updates.** We further update comprehensive supplementary materials, including **datasets, codes, requirements, commands, and log files**. Due to size limitations, we upload one processed dataset, UCI. Other datasets can be found in DyGLib (https://github.com/yule-BUAA/DyGLib) and are processed in the same way.
> > >
> > > **How to run.** To run the code, you can first install the necessary environment using `pip install -r requirements.txt`, and then execute the codes directly with `bash example.sh`. Please note that slight variations may occur between the results you obtain and the reported results due to factors such as initialization and random seeds across devices. Therefore, for your reference, we provide the log files of our reported results in `./logs/`. Despite these variations, we guarantee that the produced results are consistently and obviously higher than those of the best baseline models. Please feel free to verify them.
> > >
> > > Thank you once again for recognizing our work. Your insightful comments have played a significant role in enhancing our work!

---

> > > > ### Comment · Reviewer_AAW9 · 2024-11-28
> > > >
> > > > Thank you for the updates. I find it unusual that each node in a batch uses a unique `hidden` and `cell` in the LSTM module. Have you tried an alternative approach where all nodes share the same `hidden` and `cell`? If so, how does the performance compare? Additionally, how do you test the MRR scores with 100 negative samples, given that your implementation includes only 200 `hidden` and `cell`?

---

> ### Author Response · Authors · 2024-11-28
> **# Response for Reviewer AAW9 for the Codes**
>
> Thank you for considering our work! We are happy to provide further clarification based on your additional question.
>
> As mentioned in **Lines 912–914**, for batch processing, we maintain the rhythm vector in a batch with dimensions corresponding to the batch size, which is dynamically updated during training. Simply setting the batch size to 1 could train the model where all nodes share the same rhythm state, as the interaction is inputted one by one. However, we did not conduct this experiment because setting the batch size to 1 would significantly slow down the training process --- at least **200 times slower than the current setup!**
>
> For 100 negative nodes, we modify ``get_link_prediction_metrics()`` in the DyGLib. For each batch (suppose the batch size is 200), we select **the first 100** negative samples from 200 generated negative samples as the final negative samples for the MRR metric. We believe this method could achieve the purpose of the negative sampling strategy during the evaluation phase. The pseudocode is as follows:
>
> ``1. y_pred_pos, y_pred_neg = predicts[:num_pos], predicts[num_pos:]``
>
> ``2. y_pred_neg = y_pred_neg[:100]``
>
> ``3. computing MRR...``
>
> We are deeply grateful for your invaluable comments, which have significantly improved our paper. Your comprehensive review has been incredibly insightful.

---

> > ### Comment · Reviewer_AAW9 · 2024-11-29
> >
> > Thank you for your thoughtful clarification. However, I remain unconvinced by your explanation.
> >
> > First, I believe that a machine learning model should be independent of its training batch size. Otherwise, how can you ensure the model performs well across different batch sizes? Furthermore, the size of the model parameters increases with the training batch size, which is impractical. Moreover, in your current setup, the test batch size must match the training batch size, which introduces an unnecessary constraint. To address the issue of nodes sharing the same rhythm, there is no need to set the training batch size to 1; simply using the same rhythm for all nodes within a batch would suffice.
> >
> > Second, I disagree with your method for computing the MRR score. To correctly evaluate the MRR score for a source node, one must compute the probabilities of the source node and multiple negative samples. In other words, the MRR score should reflect the expected rank of the positive sample among all  $n-1$ samples, where  $n$  is the total number of nodes in the graph. However, the MRR score you computed is based on the rank of the positive sample among all  $n^2 - 1$  samples, since the source node is not fixed. These two settings have entirely different meanings, and your current approach does not align with the standard definition of MRR.

---

> ### Author Response · Authors · 2024-11-29
> **Response for Reviewer AAW9 for the Codes [Part 2]**
>
> We are deeply grateful for your invaluable comments! We would like to provide further clarification as follows:
>
> As mentioned in **Lines 1009-1040**, we have evaluated and discussed the model performance of TG-Mixer and existing TGNs under different batch sizes. Empirically, the performance of most existing TGNs is influenced to varying degrees by batch size. This is because all predictions within a given batch share the same model state and memory information, which may become outdated for later interactions in the batch. Consequently, the model training is more likely to experience greater information loss when using a larger batch size. For further details, please refer to Lines 1009-1040.
>
> We would like to clarify that **our computation of the MRR metric is reasonable and aligns with practices commonly used in recent works and their official implementations.** We follow the TGB benchmark [1, 2] and reference their official implementations [url1, url2, url3] to evaluate model performance using MRR for temporal link prediction. Specifically, for each positive sample, we calculate its ranking among 100 negative samples to compute the MRR. **For efficiency, all positive samples within a given batch share the same 100 negative samples.** Therefore, in the provided pseudocode, for all positive samples in a batch, i.e., ``y_pred_pos``, we sample 100 negative samples, i.e., ``y_pred_neg[:100]``. We believe this computation is both efficient and reasonable because it satisfies the randomness requirement for negative sampling while avoiding the significant complexity that arises from computing 100 negative samples for every positive sample.
>
> Thank you for your attention to our work; your feedback has been extremely helpful in improving our paper!
>
> [1] DTGB: A Comprehensive Benchmark for Dynamic Text-Attributed Graphs, NeurIPS, 2024.
> [2] Temporal Graph Benchmark for Machine Learning on Temporal Graphs, NeurIPS, 2023.
> [url1] https://github.com/zjs123/DTGB/blob/main/evaluate_models_utils.py#L157
> [url2] https://github.com/shenyangHuang/TGB/blob/main/tgb/linkproppred/evaluate.py#L76
> [url3] https://github.com/shenyangHuang/TGB/blob/main/examples/linkproppred/tkgl-wikidata/tlogic.py#L75

---

> > ### Comment · Reviewer_AAW9 · 2024-11-30
> >
> > Thank you for your response. However, I must respectfully express my complete disagreement with your comments.
> >
> > 1.	“This is because all predictions within a given batch share the same model state and memory information, which may become outdated for later interactions in the batch.”
> > Batch size affects only memory-based models, such as TGN, JODIE, and DyRep, for the reason you mentioned. However, other methods, including GraphMixer, DyGFormer, and your proposed method, utilize up-to-date historical neighbors and should not suffer from information loss. Figure 8 shows clear performance degradation for TG-Mixer and TGN only, which raises further doubts about the performance of TG-Mixer as the batch size continues to grow.
> > In my previous comments, I also proposed a solution to address my concern about the model’s reliance on batch size: “There is no need to set the training batch size to 1; simply using the same rhythm for all nodes within a batch would suffice.” I do not think evaluating the performance under various batch sizes is sufficient to address this concern.
> >
> > 2.	“We follow the TGB benchmark [1, 2].”
> > I believe your interpretation of [1, 2] is incorrect. For a positive sample (a, b), TGB computes the rank of (a, b) among  \{(a, b), (a, n_1), (a, n_2), \dots, (a, n_{100})\} . In contrast, your implementation computes the rank among  $\{(a, b), (c_1, n_1), (c_2, n_2), \dots, (c_{100}, n_{100})\}$. The MRR metric is designed to align with real-world applications. In recommendation scenarios, the goal is to recommend items to a specific user that the user is most likely to interact with. To evaluate a model’s ability to predict a user’s preferences, the positive item should be ranked highly among other negative items for that user. Therefore, it is essential to compute the probabilities for a fixed source node and multiple destination nodes. I recommend referring to the [TGB paper](https://arxiv.org/pdf/2307.01026), page 5, paragraph “Performance metric,” for clarity on this matter.

---

> ### Author Response · Authors · 2024-12-02
> **# Response for Reviewer AAW9 for Codes [Part 3]**
>
> We sincerely appreciate your meticulous attention and valuable suggestions, which have significantly made our work more thorough and rigorous.
>
> Based on your invaluable feedback, we conduct additional empirical evaluations that confirm the following two findings, and we will incorporate these discussions into the revised version of our paper.
>
> > Finding 1: **The batch-based rhythm vector could serve as an effective technique to mitigate the outdated information issue in existing batch training approach.**
>
> As mentioned in Line 1019, we would like to clarify that TG-Mixer is a memory-based method because it requires maintaining an up-to-date rhythm vector component, which is updated at the batch level.
>
> **Empirical Analysis.** As stated in Lines 1016–1018, a limitation of the existing training approach in TGNs is that all predictions within a given batch share the same model state, which may become outdated for later interactions in the batch. We believe that **the batch-based rhythm vector offers an effective solution to this issue by maintaining separate rhythm vectors for each prediction within the same batch**. This enables TG-Mixer to capture the temporal differences between interactions of a batch, thereby mitigating the adverse effects of outdated information during training.
>
> **Experimental Evaluation.** To assess the effectiveness of the batch-based rhythm vector, we conduct an additional ablation study. Specifically, we use the same rhythm vector for all nodes within a batch and update it globally. This results in a new variant, and we call it "TG-Mixer$_{\text{w/o. batch rhythm}}$".
> The experimental results are represented in Table 1, and we will include them in Table 5 of our paper. We observe **a performance drop for TG-Mixer when all nodes within a batch share the same rhythm vector**. This highlights the effectiveness of the batch-based rhythm vector, which successfully mitigates the outdated information issue inherent in the existing batch training approach.
>
> Table 1: Transductive AP (\%) results of ablation study.
>
> |Variants|Wikipedia|Reddit|UCI|Contact|
> | :-:|:-:|:-:|:-:|:-:|
> |TG-Mixer|**99.83**|**99.89**|**96.56**|**99.58**|
> |TG-Mixer$_{\text{w/o. batch rhythm}}$|98.78|98.32|95.53|97.39|
>
> > Finding 2: **TG-Mixer consistently demonstrates superior performance on the MRR metric when computed with fixed source nodes.**
>
> **Empirical Analysis.** In our implementation, we compute MRR by ranking a specific positive edge among 100 complete negative edges, where the negative source and destination nodes may differ from the positive ones. We believe this approach strikes a balance between fair evaluation and efficiency. On the other hand, in real-world applications such as recommendation systems, the effectiveness of recommending items for a given user is often of primary concern. Therefore, we conduct an additional evaluation using the MRR metric, where **the source node (i.e., user) is fixed**, and 100 negative edges are constructed by sampling 100 negative destination nodes (i.e., potential recommended items) for this source node.
>
> **Experimental Evaluation.** The experimental results are presented in Table 2, and we plan to add a new section in our paper to incorporate this discussion accordingly.
>
> TG-Mixer **continues to achieve the best performance** under the MRR metric when the source nodes are fixed, further demonstrating its effectiveness and superiority. Notably, when combined with the results reported in our paper, TG-Mixer achieves the best performance across AP, ROC-AUC, and the MRR metrics for fixed and non-fixed source nodes, highlighting its robustness and overall effectiveness.
>
> We sincerely appreciate your meaningful feedback, which has been instrumental in improving our work. Thank you once again!
>
> Table 2: Transductive MRR results with fixed source nodes.
>
> ||JODIE|TGAT|TGN|GraphMixer|DyGFormer|TG-Mixer|
> |:-:|:-:|:-:|:-:|:-:|:-:|:-:|
> |UCI|52.91|63.86|66.18|66.51|59.68|**68.37**|
> |US Legis.|44.83|48.29|47.28|63.91|70.81|**80.74**|

---

> > ### Comment · Reviewer_AAW9 · 2024-12-02
> >
> > Thank you for the additional experiments. However, regarding your Finding 1, I remain unconvinced that batch-based rhythm vectors improve model performance by mitigating the adverse effects of outdated information. The global rhythm, which is updated using the last time of the previous batch, could be more up-to-date than batch-based rhythms. I believe this is an interesting question that warrants further investigation, and I encourage the authors to explore the precise reason behind this observation and to design a new model that is independent of batch size.
> >
> > For Finding 2, how do you compute the MRR metric given the limitation of batch-dependent rhythms? I believe the new setting with fixed source nodes is the only correct approach, and the previous results should not be included in your revision.

---

> ### Author Response · Authors · 2024-12-02
> **Response for Reviewer AAW9 for Codes [Part 4]**
>
> We sincerely appreciate your thoughtful attention to our work. Regarding the concerns you raised, we would like to offer the following clarifications:
>
> > **For Finding 1.**
>
> We believe that the conclusion of Finding 1 is reasonable and well-supported. **Temporal differences across different batches (e.g., information from the previous batch) are addressed through the model update via Back Propagation Through Time (BPTT).** On the other hand, the outdated information issue we discuss pertains to outdated information **within the same batch**. The fundamental reason for this problem is that all predictions within the batch share the same model state and the information for the later predictions is outdated. Therefore, we design distinct rhythm vectors to capture temporal differences among different predictions within a given batch. This approach prevents one shared rhythm vector for all predictions within a batch, thus effectively avoiding the outdated information issue from the later predictions of the batch.
>
> > **For Finding 2.**
>
> We provide the detailed computation process for MRR with fixed source nodes as follows:
>
> **Step 1: Computing positive links probabilities.** This process is consistent with our implementation. For clarity, we describe it as ``pos_probs = TGMixer(pos_src_nodes, pos_dst_nodes, interact_times)``.
>
> **Step 2: Computing negative links probabilities.** We follow the implementation in [1] and sample 100 negative destination nodes for each batch. The source nodes within a batch share the same set of 100 negative destination nodes. The detailed implementation for each batch is as follows (Reference: [url1]):
>
> ****
> 1. Sample 100 negative destination nodes, resulting in ``neg_dst_nodes``.
> 2. Initialize ``all_neg_probs = []``.
> 3. For each``one_neg_dst_node`` in ``neg_dst_nodes:``
> 	* Duplicate ``one_neg_dst_node`` ``batch_size`` times, resulting in ``one_neg_dst_nodes``.
> 	* Compute the negative link probabilities using ``neg_probs = TGMixer(pos_src_nodes, one_neg_dst_nodes, interact_times)``.
> 	* Append ``neg_probs`` to ``all_neg_probs``.
>
> ****
> This process yields the negative link probabilities for all positive source nodes corresponding to the 100 negative destination nodes.
>
> **Step 3: Computing MRR.** For each positive link probability in ``pos_probs``, we retrieve the corresponding pos_src_node probabilities from ``all_neg_probs`` for the 100 negative links, and finally compute the MRR.
>
>
> Finally, we acknowledge that MRR with fixed source nodes is more aligned with recommendation systems. However, we believe that the MRR results with non-fixed nodes presented in our paper at least provide a fair testbed for performance evaluation, as all models are evaluated using the same metric computation method. In future versions, we will clearly document our MRR computation methods and revise the results to include MRR with fixed source nodes. Thanks for your advice!
>
> [1] DTGB: A Comprehensive Benchmark for Dynamic Text-Attributed Graphs, NeurIPS, 2024.
> [url1] https://github.com/zjs123/DTGB/blob/main/evaluate_models_utils.py#L218

---

> > ### Comment · Reviewer_AAW9 · 2024-12-03
> >
> > Thank you for the clarifications. However, I am still unclear about your claim that a single rhythm incurs an information loss problem. Suppose the timestamps of the samples in the first batch are  $t_1, \ldots, t_{200}$ , and the timestamps in the second batch are  $t_{201}, \ldots, t_{400}$ . A single rhythm will contain up-to-date information at  $t_{200}$ , as it is updated with information from  $t_1$  to  $t_{200}$ . In contrast, the 200 batch-based rhythms will reflect information from  $t_1, \ldots, t_{200}$  individually. Why are the batch-based rhythms considered more up-to-date than a single rhythm?
> >
> > Regarding the MRR setting, I believe using the same negative samples for different source nodes introduces biases in the evaluation. I encourage adopting the MRR setting used in TGB, where destination nodes are sampled randomly for each source node. Moreover, using a fixed source node not only aligns better with recommendation scenarios, but is also standard practice in the dynamic graph learning literature. This is because the MRR metric typically evaluates a single user’s preferences, making fixed source nodes essential for meaningful evaluation. Therefore, I strongly recommend avoiding the use of different source nodes for MRR evaluation. To address efficiency concerns, you could only adopt MRR as the test metric while relying on AP for validation. The MRR test would only take a few hours to complete.

---

> ### Author Response · Authors · 2024-12-03
> **Response for Reviewer AAW9 for Codes [Part 5]**
>
> Thank you for your attention. Based on your further question, we provide the following response:
>
> We would like to emphasize that **the single rhythm vector is updated recursively in a per-sample manner.** This means that the single rhythm vector is progressively updated according to the position of predictions within the batch. For example, assuming the positions of the batch samples are $p_1, \dots, p_{200}$, the single rhythm vector is updated sequentially from position $p_1$ to $p_{200}$. Even though the interactions are already sorted by timestamp, this approach overlooks the fact that **interactions often share a large number of identical timestamps**. Assuming the interactions at positions $p_{100}$ and $p_{101}$ occur at the same timestamp, the single rhythm vector incorporates the information from the position $p_{100}$ to determine the state for $p_{101}$. This process introduces unintended temporal dependencies between these two simultaneous interactions.
> To address this issue, the batch-based rhythm vector method assigns distinct rhythm vectors to different positions within the batch, allowing **each rhythm vector to rely exclusively on the timestamp information of its corresponding position**. This approach effectively eliminates the erroneous temporal dependencies introduced by per-sample updates, ensuring an up-to-date representation of the temporal information for each interaction.
>
> Regarding the computation of MRR, we sincerely appreciate your valuable suggestions regarding efficiency. In future versions of the paper, we will revise our description of the MRR calculation and incorporate the experimental results for MRR with fixed source nodes.
>
> Finally, we would like to express our heartfelt gratitude for your insightful comments, which have greatly contributed to making our paper more rigorous!

---

> > ### Comment · Reviewer_AAW9 · 2024-12-03
> >
> > Thank you for the clarifications. I am uncertain why you do not update the single rhythm once per batch, as this approach appears more logical and would prevent the training process from becoming excessively time-intensive, akin to setting the batch size to 1.
> >
> > As the discussion period draws to a close, I would like to share my overall comments. The idea behind TG-Mixer is great, and its performance is impressive. However, the model’s dependence on batch size is unusual and limits its applicability. I recommend revising the model design to make it independent of batch size. I believe that the revised TG-Mixer could advance the field of dynamic graph learning. Wish all the best to you.

---

> > > ### Author Response · Authors · 2024-12-03
> > > **Thanks for Reviewer AAW9**
> > >
> > > We sincerely appreciate your attention to and recognition of our work. Your valuable feedback has greatly contributed to improving the quality of our research.
> > >
> > > First, we would like to address your concerns regarding the outdated information issue during training. While it is easy to extend the rhythm vector to be updated per batch, we believe that the single rhythm vector inevitably suffers from the outdated information issue, regardless of whether it is updated per batch or per interaction. (i) If updated per batch, all interactions within a given batch would share the same rhythm vector, leading to the outdated information issue for the later interactions of the same batch (see Lines 1009-1040 in our paper and "Response for Reviewer AAW9 for Codes [Part 3]"). (ii) On the other hand, if updated per interaction, unnecessary temporal dependencies may be introduced between interactions occurring simultaneously, preventing an up-to-date representation of temporal information for each interaction (see "Response for Reviewer AAW9 for Codes [Part 5]").
> > >
> > > Secondly, we deeply appreciate the effort and dedication you have invested in our work, as well as the valuable suggestions you have provided.
> > >
> > > * **Thank you for recognizing our innovative idea.** We deeply appreciate your high regard for the idea and novelty behind TG-Mixer. We are proud to be the first to explore burstiness in temporal graphs and leverage this insight to advance temporal link prediction.
> > > * **Thank you for recognizing the reproducibility of the outstanding performance in our work.** Our proposed TG-Mixer achieves state-of-the-art results in AP, AUC-ROC, and MRR across seven datasets from the DyGLib benchmark and five datasets from the TGB benchmark.
> > > * **Thank you for providing an interesting and meaningful future research direction.** We validate and discuss the performance of both existing methods and TG-Mixer under different batch sizes in Lines 1009-1040. TG-Mixer consistently outperforms the baseline models across varying batch sizes. We also find that the performance of memory-based methods (e.g., TGN and TG-Mixer) can be influenced by batch size due to the limitations of the existing batch-based training approach. From this perspective, batch size can be viewed as a hyper-parameter for memory-based methods, including our proposed TG-Mixer. Additionally, we observe that our proposed batch-based rhythm vector alleviates the limitation of the existing training approach to some extent. Addressing this limitation at the root level of the training process remains an intriguing and valuable research direction, and we agree that it warrants further investigation.
> > >
> > > Finally, we will revise our paper according to your suggestions and our corresponding responses. Thank you once again!

---

### Official Review · Reviewer_qT7Y · 2024-11-05

**Soundness:** 2
**Presentation:** 3
**Contribution:** 3
**Rating:** 6
**Confidence:** 3

**Summary:**

Authors propose the framework (TG-Mixer) that captures the bursty interactions better, given that such bursty interactions are common behaviors in the real-world datasets. Authors perform both macro and micro level analysis about the bursty interaction evidence. Based on this evidence, authors introduce the silence decay mechanism and mix it into the temporal encoding through LSTM so that the temporal encoding leverages the silence more explicitly. Based on this assumption, recent neighbor sampling strategy is proposed for the given model framework.

Authors demonstrate that the proposed method outperforms baselines in the temporal link prediction tasks. Furthermore, this performance can be achieved with less training time and fewer computational cost since the algorithm converges fast.

Finally, authors perform experiments about the generalizability of the silence decay mechanisms as well as the ablation study about proposed components, which include the silence decay mechanism and recent neighbor sampling. The ablation study proves the importance of the proposed components.

**Strengths:**

- The introduction of the silence decay mechanism makes sense and works well. This component seems to deliver the inactivity information more effectively than temporal models (e.g. LSTM) learn from data.
- The proposed method shows superior performance as opposed to the baseline models.
- Ablation study helps the understanding of the roles for each proposed component.

**Weaknesses:**

- The analysis about the silence duration (Figure 2) is not fair to compare because many of random pairs would not happen forever in the existing datasets. While the low silence duration for positive links makes sense, it is not clear to understand whether high silence duration in the random selection is mostly originated from the pairs that would never link or from the inactive period of the potentially interacting pairs.

- Similarly, the discriminative signal argument in 5.3 is overstated by the reason mentioned in the previous bullet point. The following phenomena can explain this as well: (1) if the pairs have never interacted before, then they are also unlikely to interact in the future. (2) if the pairs recently interacted, then they are likely to interact in the current timeframe. Temporal clustering should represent burstiness or not (e.g. periodic interactions), but no interactions are mixed in the analysis.

**Questions:**

- The idea of the silence decay seems irrelevant to graph data. Have the authors applied this idea into the other bursty time-series prediction? Or if this idea should be related to GNN (i.e. neighbor samples), what is the main reason for that?

- Does the ablation study cover where g(s) = 0 case? That is, the proposed framework works with LSTM and everything but the silence decay, and it would be good to check its performance as well as convergence. This ablation study would be interesting since LSTM is supposed to capture the temporal patterns but the silence decay provides some short-cut to the important information. Hence, it would be great to understand how temporal mix can suffer without this shortcut.

- Silence decay component may clearly discriminates easy negative links more efficiently. It would be great to check the performance on brand-new links (new links that never interacted before) or resurrecting links (interactions after some inactive period)

- This idea would seem to work well for the periodic interactions (e.g. location check-ins). It would be great to show this empirically. Otherwise, the proposed method would be limited only to burst interaction networks.

---

> ### Author Response · Authors · 2024-11-20
> **Response to Reviewer qT7Y [Part 1]**
>
> We would like to sincerely thank Reviewer qT7Y for providing a detailed review with insightful questions.
>
> > **W1:** Explanation of the conclusion for empirical analyses.
>
> *Quick Answer: We would like to clarify that our empirical analyses are fair because they are specifically **designed to compare the differences between truly existing links and links that would never occur.***
>
> Our empirical analyses do not consider the randomly sampled interaction nodes as "potentially interacting pairs". Instead, they represent "the pairs that would never link" as you noted. As outlined in Lines 157-158, our empirical analyses compare truly existing interaction nodes with randomly sampled ones. **This setup aligns directly with the link prediction task**, where the goal is to distinguish positive links (those that truly exist) from negative links (those that do not). Therefore, the conclusions drawn from our empirical analyses are both reasonable and well-supported.
>
> > **W2:** Explanation of bursty and periodic interactions.
>
> *Quick Answer: We do not differentiate between bursty and periodic interactions in our analyses because **both of them exhibit temporal clustering behavior.***
>
> **Empirical Analysis.** Both bursty and periodic interactions exhibit temporal clustering, with the key distinction being whether their timestamps follow a regular pattern.
>
> * Bursty interactions are characterized by unevenly distributed timestamps, and typically demonstrate temporal clustering **during specific periods**.
> * Periodic interactions, in contrast, feature regular timestamps but can also demonstrate temporal clustering **at a specific timestamp**, as a large number of interactions may occur simultaneously at this timestamp.
>
> Since our focus is not on the regularity of timestamps, we do not separately analyze these two types of interactions.
>
> **Experimental Evaluation.** As noted in Q4, we have incorporated periodic datasets into the model evaluation. TG-Mixer significantly outperforms the baseline models on these periodic datasets, further demonstrating its effectiveness. For additional details, please refer to Q4.
> ****
> **Thanks for your insightful review, which has guided our revisions!** As revised in Lines 242-245, we elaborate on temporal clustering for temporal link prediction based on your insightful review in W2. Temporal clustering can be explained as follows:
>
> * (i) If a given node has recent interactions, then it is more likely to interact at the current timestamp;
> * (ii) If the node has only distant past interactions or even no historical interactions, it is less likely to interact now.
>
> > **Q1-1:** Comparison with existing burstiness study.
>
> As revised in Lines 1306-1314, we have identified some existing studies on burstiness that explore the effects of bursty interaction patterns. However, these related works focus on physical science and **outside the ML community**, and we have not found relevant studies within the ML community.
>
> **Difference Comparison.** Although there indeed exist some works that explore burstiness, these works focus on designing statistical models to analyze bursty events and their impacts on the distribution characteristics, correlations, and propagation processes. They are outside the ML community. Instead, we present the first attempt to investigate temporal clustering in temporal graphs, and incorporate this insight for advancing temporal link prediction.
>
> > **Q1-2:** Explanation of motivation for silence decay in temporal graphs.
>
> *Quick Answer: We would like to clarify that the silence decay mechanism may not be directly relevant to high-order neighbors in graph data but is **highly relevant to first-hop neighbors in temporal graphs**.*
>
> * **We investigate the activity patterns in temporal graphs and study invaluable insights for temporal link prediction.** As mentioned in Lines 246-251, the silence decay mechanism stems from the implications of empirical analyses. As shown in Figure 1 of Line 66, in temporal graphs, the first-hop neighbors of specific nodes can be represented as an interaction sequence, which reveals their activity patterns. As illustrated in Line 243, we study such patterns and look for potential insights for temporal link prediction.
> * **Motivation for silence decay.** As detailed in Figure 3 of Line 181, inter-event times play a crucial role in incorporating the powerful node interaction rhythms. To this end, we designed the silence decay mechanism to reflect current rhythm patterns, promoting the inclusion of temporal clustering information for temporal link prediction.

---

> > ### Author Response · Authors · 2024-11-20
> > **Response to Reviewer qT7Y [Part 2]**
> >
> > > **Q2:** Ablation study for silence decay mechanism.
> >
> > *Quick Answer: TG-Mixer **experiences a significant performance decrease when the silence decay mechanism is removed.***
> >
> > *Revision: As revised in Lines 520-529, we have conducted an ablation study for the silence decay mechanism.*
> >
> > **Empirical Analysis.** To evaluate the effectiveness of the silence decay mechanism, we remove this module in TG-Mixer by setting the decay coefficient $\text{g}(\cdot) = 0$ of Equation 3, leading to an additional ablation variant "TG-Mixer$_{\text{g}(\cdot)=0}$". Due to char limitations, we provide a subset of the results below. For the complete results, please refer to Tables 4 and 14 in our revised paper.
> >
> > **Experimental Evaluation.** From the results, We see that TG-Mixer suffers a significant performance drop when the silence decay mechanism is removed. This further demonstrates the critical role of capturing temporal clustering information for temporal link prediction.
> >
> > Table 1: Transductive AP (\%) results of ablation study.
> >
> > |Variants|Wikipedia|Reddit|UCI|Contact|
> > | :-:|:-:|:-:|:-:|:-:|
> > |TG-Mixer|**99.83**|**99.89**|**96.56**|**99.58**|
> > |TG-Mixer$_{\text{g}(\cdot)=0}$|97.34|96.10|93.91|94.81|
> >
> > > **Q3:** Evaluation under different negative sample strategies.
> >
> > *Quick Answer: TG-Mixer **demonstrates outstanding performance across four challenging negative sampling strategies** under the MRR metric, reinforcing its effectiveness.*
> >
> > *Revision: We have incorporated these discussions in Lines 972-1007.*
> >
> > **Empirical Analysis.** As stated in Line 908, We evaluate model performance using the random sampling strategy. Recently, some works [1, 2] have introduced advanced evaluation methods for temporal link prediction in temporal graphs. To fully assess the models, we adopt a more challenging evaluation with four negative sampling strategies and rank-based metrics.
> >
> > **Experimental Evaluation.** Based on the above motivation, we conduct additional experiments to evaluate model performance under four different negative sampling strategies using the MRR metric. Their descriptions can be found in the Implementation Details.
> >
> > Due to char limitations, we provide key findings and results here. Other findings can be found in Lines 972-1007.
> >
> > * TG-Mixer **consistently performs best** across four negative sampling strategies under the MRR metric, further proving its robustness.
> > * Inductive negative sampling often leads to the most performance drop. This is likely because this strategy samples unseen nodes to construct negative links, making it more challenging for models to distinguish between positive and negative links.
> >
> > Table 1: A subset of transductive MRR results on Reddit.
> >
> > ||JODIE|TGAT|TGN|DyGFormer|TG-Mixer|
> > |:-:|:-:|:-:|:-:|:-:|:-:|
> > |random|70.24|64.24|72.95|81.47|**88.10**|
> > |historical|61.83|60.77|62.81|70.16| **73.80**|
> > |inductive|63.18|55.18|57.94|68.18| **70.51**|
> > |degree-aware|68.94|59.17|60.19|71.92|**74.24**|
> >
> > **Implementation Details.** In addition to random negative sampling, we follow [1] and use **historical negative sampling** (sampling negative links that have been observed) and **inductive negative sampling** (sampling negative links that are unseen during training). To mitigate potential biases from popular nodes, we conduct **degree-aware negative sampling** by sampling negative nodes with the probability of $d/N$, where $d$ is the node's current degree and $N$ is # historical interactions. We follow [2] and sample 100 negative edges per positive edge and employ MRR as the metric.
> >
> > [1] Towards Better Evaluation for Dynamic Link Prediction, NeurIPS, 2022.
> > [2] Temporal Graph Benchmark for Machine Learning on Temporal Graphs, NeurIPS, 2023.
> >
> > > **Q4:** Analysis for periodic interactions.
> >
> > *Quick Answer: Some datasets used in our paper inherently feature periodic interactions, and TG-Mixer **exhibits exceptional performance when handling periodic interactions.***
> >
> > * **Inclusion of periodic interactions.** Table 6 of Line 838 summarizes the statistics and details of the datasets. We observe that the 'US Legis.' and 'Flights' datasets exhibit regular timestamps and can be regarded as examples of periodic interactions. As analyzed in Line 1412 of our paper and W2, these datasets exhibit a large number of simultaneous interactions, which amplify temporal clustering patterns.
> > * **Experimental Evaluation.** The experimental results in Table 1 of Line 324 demonstrate that TG-Mixer significantly outperforms baseline models on the US Legis. and the Flights datasets, underscoring its effectiveness in handling periodic interactions.
> >
> > Again, we express our gratitude for your suggestions, and we hope our revised version has addressed your problems.

---

> ### Author Response · Authors · 2024-12-03
> **Thanks for your positive reviews!**
>
> We are sincerely grateful for your positive assessment! We are deeply grateful for your recognition and suggestions, and we have revised our paper based on your reviews.
>
> Once again, many thanks for your insightful efforts. Your detailed suggestions help us a lot!

---

### Official Review · Reviewer_nfv5 · 2024-11-06

**Soundness:** 2
**Presentation:** 2
**Contribution:** 2
**Rating:** 3
**Confidence:** 4

**Summary:**

The authors of this work propose the temporal graph learning model TG-Mixer, which is based on an encoder-decoder architecture. Building on the observation that nodes in many temporal networks exhibit bursty interaction patterns, the encoder uses a sample of recently occurring interactions for a given node u and timestamp t, mixing a concatenation of neighbor and link features with a vector-based representation of temporal interaction patterns that are fed into an LSTM model. Addressing a dynamic link prediction task, the proposed architecture is experimentally evaluated against multiple baseline models, showing good performance in seven empirical temporal network data sets. An additional analysis of training times shows that the model can be trained more efficiently than the competing baseline methods. Additional results in the appendix consider standard benchmark graphs from the TGB community benchmark, an ablation study that replaces the information mixer with an attention mechanism, as well as additional experiments on partially shuffled data sets that partially destroy temporal clustering.

**Strengths:**

[S1] The authors propose a novel temporal graph learning architecture to address the temporal link prediction problem, which is an important problem in many scenarios.

[S2] The proposed method allows to incorporate both temporal and topological patterns in temporal graphs.

[S3] The authors performed a rather extensive set of experiments, with many different baseline models, several empirical data sets, as well as an ablation study and experiments on partially shuffled data sets.

[S4] The results of the experimental evaluation are promising, showing (mild) increases of performance against many baseline models as well as considerably reduced training times.

**Weaknesses:**

Unfortunately, I also found several weaknesses, which I briefly list in the following. For each of those weaknesses, I include specific and more detailed questions to the authors below:

[W1] The claimed contribution of observing bursty interaction patterns in temporal graphs is not new and the work should be better aligned with the large body of existing literature on bursty interaction patterns in temporal graphs.

[W2] Key components of the TG-Mixer encoder, which is a central component, are not explained well.

[W3] The problem formulation is unclear and does not seem to match the empirical evaluation.

[W4] The authors did not properly account for recently highlighted challenges in the evaluation of dynamic link prediction.

**Questions:**

Regarding [W1], in the introduction the authors claim that one of the two key contributions of this work is that they "observe a prevalent and strong pattern of temporal clustering in node interaction rhythms". However, this is hardly an original contribution of this work, as there are dozens of works outside the ML community that have not only observed but also quantitatively characterized and modeled bursty interaction patterns in temporal networks. A very incomplete selection of works published over the past ten years include:

- M Karsai et al.: Small but slow world: How network topology and burstiness slow down spreading, PRE, 2011
- R Lambiotte et al.: Burstiness and spreading on temporal networks, EPJ, 2013
- T Hiraoka: Modeling temporal networks with bursty activity patterns of nodes and links, Phys Rev. Research 2, 2020
- B Min and KI Goh: Burstiness: Measures, models, and dynamic consequences, Temporal Networks, 2013
- M Karsai et al.: Empirical findings in human bursty dynamics, Bursy Human Dynamics, 2018
- Hang-Hyun Jo et al.: Bursty Time Series Analysis for Temporal Networks, Temporal Network Theory, 2023

Related to this missing related work, it is not ideal that the authors do not use terminology that is common in the field. For instance, they renamed the concept of inter-event times, which is common both in temporal networks and time series analysis, "silent duration". This makes the work unnecessarily hard to read.

Finally, it is a well-known fact that bursty interaction patterns are associated with non-Poissonian, heavy-tailed inter-event time distributions, for which the mean is not a meaningful characterization. As such, the analysis of mean silence durations in Figure 2 is not insightful and the resulting comparison to randomly sampled interactions simply repeats the well-known fact that the distributions are not Poissonian (as expected from a memoryless, i.e. Markovian process).

Considering [W2] from the description in section 4 I could neither understand the definition nor the motivation of the TG-Mixer Encoder component. In particular, how exactly is $C_{rhythm}^t$ defined? How is the information on "interaction rhythms" encoded in the "communal rhythm vector"? What is the motivation for the definitions of $C_{temp}^t$ and $C_{decay}^t$?

Regarding [W3] I have concerns regarding the problem formulation, especially whether it actually matches the task addressed in the experimental evaluation. In Definition 2, the authors state that they address the task of predicting the presence of a link between a specific pair of nodes and at a *specific* timestamp t. Notably, this is more difficult than predicting whether a link will exist within a given time interval (or batch) or whether a link will exist in the future, i.e. *after* a given timestamp t. In the description of the experimental evaluation, I did not find further details of how this specific link prediction task has been evaluated. My assumption is that the authors have rather addressed a batch-based prediction, i.e. predicting whether a link occurs in a given batch in the test/validation set. Whether or not this task corresponds to the task specified in Definition 2 depends on (i) the batch size, and (ii) the characteristics of the temporal network (i.e. whether it a discrete- or continuous-time temporal graph). Unfortunately no details on batch sizes are mentioned in the description of the experimental results, which would be crucially important for interpretation. Moreover, seeing the very high values of the average precision scores both for the proposed model and the basline methods makes me doubt that the authors actually addressed the task of predicting the presence of a link at a specifc time stamp t.

Related to weakness [W4], I would like to refer to the challenges in the evaluation of dynamic link prediction that have been outlined in:

Poursafaie et al: Towards better Evaluation for Dynamic Link Prediction, NeurIPS, 2022

This work introduced EdgeBank and is cited in the paper for this reason. However, it also highlights the need to carefully consider different negative sampling strategies in the evaluation of dynamic link prediction, which have been shown to considerably influence the performance (and ranking) of temporal graph learning models - even for some of the data sets used in the present work. Since the authors cite this work, it is unclear why only the simplest negative sampling method has been used in the experimental evaluation. Also some of the other proposed negative sampling methods are implemented in the temporal graph learning library used by the authors, which would have made it easy to include those as additional results (I also checked but did not find those in the appendix).

Moreover, I appreciated that the authors followed the suggestion of the temporal graph benchmark TGB, adopting a rank-based evaluation in terms of the mean reciprocal rank (MRR) in the supplemental experimental results in the appendix. However, I think that all of the results on the five TGB benchmark data sets for dynamic link property prediction should be shown in the paper - not only those two for which - according to the authors - the proposed method ranks first and third. FInally, I would kindly ask to include the rank-based MRR scores also for the data sets considered in the main paper.

Due to the open issues outlined above, I currently evaluated this work with a score of 3, but would be open to revise this judgment if my concerns are addressed adequately.

---

> ### Author Response · Authors · 2024-11-20
> **Response to Reviewer nfv5 [Part 1]**
>
> We sincerely thank Reviewer nfv5 for providing the insightful review, and we have revised our paper based on your review carefully.
> > **W1-1:** Difference discussion and contribution reaffirmation.
>
> *Quick Answer: Unlike the existing burstiness studies in the physical science community, we present the **first model** that incorporates burstiness for temporal graph representation learning.*
>
> *Revision: For readability, we have updated the term from "silence duration" to "inter-event times" throughout the paper and developed a discussion on existing burstiness studies in Lines 1303–1311.*
>
> **Existing Burstiness Studies.** As revised in Line 1301, although some works explore burstiness in sequential data, they focus on designing **statistical models** to analyze bursty events and their impacts on distribution, correlations, and propagation. These are outside the ML community.
>
> **Our Work.** Instead, we leverage burstiness to design a **deep-learning model** tailored for representation learning in temporal graphs, and we highlight our key contributions:
>
> * **Empirical analyses for temporal clustering.** We introduce statistical empirical analyses and present the first attempt to identify temporal clustering in temporal graphs, i.e., the activity of certain nodes exhibits burstiness.
> * **A novel model for temporal link prediction.** Building on the insights from empirical analyses, we propose the first model that explicitly captures burstiness information for temporal link prediction.
>
> > **W1-2:** Explanation of mean operation in empirical analyses.
>
> *Quick Answer: We would like to clarify that the **mean operation is statistically valid**, and the alignment between our findings and the conclusions in existing studies further supports its reliability.*
>
> * **Mean operation is also utilized in the analysis of existing studies on burstiness.** Based on the existing studies [1, 2, 3, 4], the inter-event times **for specific nodes or timestamps** exhibit a heavy-tailed distribution. The mean operation in empirical analyses derives the inter-event times for each node or timestamp, which is also conducted in existing studies. For example, in Eq. 1 of [1] and Fig. 1 of [2], the authors compute the average inter-event times for each node to model the heavy-tailed distribution.
>
> * **Conclusion reliability.** As shown in Figure 3, the inter-event times for truly existing interaction nodes follow a typical heavy-tailed distribution. This finding is consistent with the conclusion in existing studies [1, 2, 3, 4], which further supports the validity of analyses.
>
> [1] Modeling temporal networks with bursty activity patterns of nodes and links, Phys Rev. Research 2, 2020.
> [2] Small But Slow World: How Network Topology and Burstiness Slow Down Spreading, PRE, 2011.
> [3] Constructing temporal networks with bursty activity patterns, Nature Communications, 2023.
> [4] Copula-based Analysis of the Autocorrelation Function for Simple Temporal Networks, KPS, 2023.
> > **W2-1:** Explanation of the rhythm vector.
>
> *Revision: In Lines 285-347, we have revised and rewritten the description of the TG-Mixer encoder.*
>
> **Motivation Analysis.** As mentioned in Lines 303-310, we introduce the rhythm vector $\mathbf{C}_\text{rhythm}^t$ to encapsulate the condensed essence of interaction rhythms at each timestamp. This vector is updated globally and chronologically through the temporal mixer. Intuitively, the interaction rhythm for the next timestamp is influenced not only **by the current rhythm patterns (achieved by the silence decay mechanism)** but also **by the historical rhythms among recent interactions (achieved by the information mixer).**
>
> **Update Process.** Based on this motivation, **as clearly shown in the "Temporal Mixer" part of Figure 4**, the update process of the rhythm vector consists of two steps: (i) a silence decay mechanism that captures current rhythms through inter-event times (Eq. 3, 4), and (ii) the information mixer, which updates the rhythm vector with historical rhythms among recent interactions (Eq. 5).
> > **W2-2:** Motivation of the $\mathbf{C}_ {\text{temp}}^t$ and $\mathbf{C}_ {\text{decay}}^t$.
>
> *Revision: In Lines 311-323, we have revised the description of the silence decay mechanism.*
>
> As noted in Line 311, the silence decay mechanism aims to update the rhythm vector with current rhythm patterns. In Line 320, we have mentioned that the rhythm vector records the long-term state because it is frequently updated across timestamps. Therefore, in our silence decay mechanism, we first generate **short-term rhythm state** $\mathbf{C}_ {\text{temp}}^t$ using a neural network. Then, the long-term rhythms are decayed based on the inter-event times, thus capturing and encoding temporal clustering patterns among current rhythms. This results in the decayed vector $\mathbf{C}_ {\text{decay}}^t$, which **contains the current rhythm pattern information**. More details can be found in Lines 311-323 of our revised paper.

---

> ### Author Response · Authors · 2024-11-20
> **Response to Reviewer nfv5 [Part 2]**
>
> > **W3:** Explanation of temporal link prediction.
>
> *Quick Answer: Several sections emphasize that **the link prediction task is performed at a specific timestamp.***
>
> * **Preliminaries.** As shown in Lines 124 and 1290, temporal link prediction is a well-defined and widely studied task, which is conducted at a specific timestamp.
> * **Solution.** Line 356 states that the Temporal Link Decoder module predicts link existence at a specific timestamp.
> * **Experimental evaluation.** We follow the standard task and evaluation setting in [1]. As discussed in Line 917, we use the common setup and employ a mini-batch training with a batch size of 200. Although we conduct parallel training and inference in batches, node pairs and corresponding timestamps in each batch are completely independent. Thus, the batch size does not typically affect the performance.
>
> [1] Towards Better Dynamic Graph Learning: New Architecture and Unified Library, NeurIPS, 2023.
>
> > **W4-1:** Evaluation under four negative sampling strategies.
>
> *Quick Answer: TG-Mixer **demonstrates outstanding performance across four challenging negative sampling strategies** under the MRR metric, reinforcing its effectiveness.*
>
> *Revision: We have incorporated these discussions in Lines 972-1007.*
>
> **Empirical Analysis.** As stated in Line 908, We evaluate model performance using the random sampling strategy. Recently, some works [1, 2] have introduced advanced evaluation methods for temporal link prediction in temporal graphs. To fully assess the models, we adopt a more challenging evaluation with four negative sampling strategies and rank-based metrics.
>
> **Experimental Evaluation.** Based on the above motivation, we conduct additional experiments to evaluate model performance under four different negative sampling strategies using the MRR metric. Their descriptions can be found in the Implementation Details.
>
> Due to char limitations, we provide key findings and results here. Other findings can be found in Lines 972-1007.
>
> * TG-Mixer **consistently performs best** across four negative sampling strategies under the MRR metric, further proving its robustness.
> * Inductive negative sampling often leads to the most performance drop. This is likely because this strategy samples unseen nodes to construct negative links, making it more challenging for models to distinguish between positive and negative links.
>
> Table 1: A subset of transductive MRR results on Reddit.
>
> ||JODIE|TGAT|TGN|DyGFormer|TG-Mixer|
> |:-:|:-:|:-:|:-:|:-:|:-:|
> |random|70.24|64.24|72.95|81.47|**88.10**|
> |historical|61.83|60.77|62.81|70.16| **73.80**|
> |inductive|63.18|55.18|57.94|68.18| **70.51**|
> |degree-aware|68.94|59.17|60.19|71.92|**74.24**|
>
> **Implementation Details.** In addition to random negative sampling, we follow [1] and use **historical negative sampling** (sampling negative links that have been observed) and **inductive negative sampling** (sampling negative links that are unseen during training). To mitigate potential biases from popular nodes, we conduct **degree-aware negative sampling** by sampling negative nodes with the probability of $d/N$, where $d$ is the node's current degree and $N$ is # historical interactions. We follow [2] and sample 100 negative edges per positive edge and employ MRR as the metric.
>
> [1] Towards Better Evaluation for Dynamic Link Prediction, NeurIPS, 2022.
> [2] Temporal Graph Benchmark for Machine Learning on Temporal Graphs, NeurIPS, 2023.
>
> > **W4-2:** Evaluation on additional datasets in TGB benchmark.
>
> *Quick Answer: TG-Mixer **exhibits superior performance across the additional datasets** in the TGB benchmark.*
>
> **Empirical Analysis.** In the original paper, we conduct experiments on the two selected datasets because many baselines may not be implemented on other datasets due to computational issues ('NA' in the following Table). To further validate the model performance, as revised in Line 956, we add three datasets: Review, Comment, and Flight.
>
> **Experimental Evaluation.** We provide a subset of experimental results here. For the complete results, please refer to Table 7 of Line 956.
>
> * TG-Mixer **consistently demonstrates superior performance** on the additional datasets, highlighting its robustness and strong capabilities.
> * The most significant performance improvements are observed in the Review and Flight datasets. This can be attributed to their higher interaction density (computed as #node degree / #time steps, as described in Line 1408), where nodes tend to feature a larger number of simultaneous interactions. Such interaction patterns result in pronounced burstiness and stronger temporal clustering, thus enhancing the effectiveness of TG-Mixer.
>
> We hope our revised paper has addressed your concerns!
>
> Table 2: A subset of MRR results on TGB.
>
> ||DyRep|TGN|GraphMixer|DyGFormer|TG-Mixer|
> |:-:|:-:|:-:|:-:|:-:|:-:|
> |Review|40.06|37.48|36.89|22.49|**49.92**|
> |Comment|NA|NA|76.17|67.03|**82.17**|
> |Flight|NA|NA|77.66|NA|**84.88**|

---

> > ### Comment · Reviewer_nfv5 · 2024-11-25
> >
> > Thank you for the responses, which however did not suceed in mitigating my concerns.
> >
> > I appreciate that the large body of works on bursty interaction patterns has now been cited (the fact that this was outside the ML community is hardly an argument!).
> >
> > I am also not convinced by the comments regading the definition of the task. I did understand that this is what the authors (and that this is common definition). However, the authors then continue arguing that they use the common batch-based approach, which exactly does not match this task definition. The fact that it is common to use this definition does not mean that it is correct. This is an issue that persists across the community (specifically in the library used by the authors, which simply gets the evaluation wrong in the sense that it does NOT match the task definition unless the batch size is chosen to be one). I thus highly doubt that the batch size does not affect the performance.

---

> ### Author Response · Authors · 2024-11-26
> **Response to Reviewer nfv5 for W3**
>
> Thank you for your thoughtful review and for recognizing the **innovation, model design, extensive experiments, and exceptional effectiveness of TG-Mixer**, as well as for acknowledging other parts of our rebuttal.
>
> We are also deeply grateful for your meticulous comments, which have helped us identify an issue within the community.
>
> > **Task definition on the batch scale.**
>
> We carefully reconsider the widely used task definition in the temporal graph community, which is typically framed at individual node pairs. In practice, however, most methods employ a batch-based training approach to enable parallel processing. Therefore, as revised in Lines 132-135, we define our task on the batch scale.
>
> **Definition 2. *Temporal Link Prediction.*** Given a batch size $B \in \mathbb{Z}^+$, a set of interaction nodes $\left\\{u_b \in \mathcal{V} \right\\}_ {b=1}^B$ and $\left\\{v_b \in \mathcal{V}\right\\}_ {b=1}^B$, and corresponding timestamps $\left\\{t_b > 0 \right\\}_ {b=1}^B$, temporal link prediction aims to predict whether each node pair $\left(u_b, v_b\right)$ interacts at timestamp $t_b$ based on the historical links $\left\\{(u', v', t') | t' < t_b\right\\} \subseteq \mathcal{G}$, i.e., predicting the existence of the future link $(u_b, v_b, t_b)$.
>
> > **Explanation of batch-based training approach.**
>
> *Quick Answer: Memory-based TGNs tend to face challenges with large batch sizes due to the outdated memory state within the batch.*
>
> *Revision: We have incorporated these discussions in Lines 1009-1040.*
>
> **Empirical Analysis.** The batch-based training approach, introduced by [1], has been widely adopted in the temporal graph community. This method first sorts all interactions by timestamp and then groups them into batches using the predefined batch size. In this way, Back Propagation Through Time (BPTT) within each batch allows the models to update in chronological order.
> Despite its efficiency, a fundamental issue with this approach is that **all predictions within a given batch share the same model state, which may be outdated for later interactions in the batch.** It is particularly problematic for memory-based methods, as they always explicitly maintain an up-to-date memory component, such as the rhythm vector in TG-Mixer and node state in TGN [1]. While the memory state for the first interaction in the batch is up-to-date (as it incorporates information from all previous interactions), the memory state for later interactions in the batch is out-of-date (as it does not include information from previous interactions within the same batch). Consequently, a larger batch size may result in more information loss.
>
> **Experimental Evaluation.** Based on the above motivation, we conduct additional experiments to evaluate model performance under different batch sizes.
>
> From the results, we find that:
>
> * TG-Mixer **continues to achieve the best performance** across various batch sizes, further validating its effectiveness and robustness.
> * Although the performance ranking of models remains unchanged, memory-based methods (e.g., TG-Mixer and TGN) tend to perform better with smaller batch sizes. This underscores the limitations of the existing batch training approach when applied to larger batch sizes. It also highlights the need for a more effective parallel processing method, which is beyond the scope of this work and we leave this as a future research direction.
>
> Once again, thanks for your valuable suggestions during both the review and discussion phases.
>
> Table 1: MRR results under different batch sizes on Wikipedia. "\*" denotes memory-based methods.
>
> ||$B=64$|$B=128$|$B=200$|$B=256$| *average ranking* |
> |:-:|:-:|:-:|:-:|:-:|:-:|
> |GraphMixer |81.30| 81.91| 80.38| 80.71| *3.0*|
> |DyGFormer|84.20| 84.16| 84.24| 84.41| *2.0*|
> |TGN\* | 74.91| 72.61| 71.98| 69.84| *4.0* |
> |TG-Mixer\*|**89.60**|**88.71**|**87.54**|**86.29**|***1.0*** |
>
> [1] Temporal Graph Networks for Deep Learning on Dynamic Graphs, ICLR, 2020.

---

> > ### Author Response · Authors · 2024-12-03
> > **Thanks for your review and the efforts for our work!**
> >
> > Thanks for your constructive comments! Your constructive reviews, especially regarding the limitation of the existing batch-based training approach, inspire us a lot. Your detailed suggestions have been immensely helpful, and we have revised our paper carefully based on your reviews.
> >
> > Once again, thank you for your dedicated efforts!

---

### Official Review · Reviewer_S1Kp · 2024-11-07

**Soundness:** 3
**Presentation:** 4
**Contribution:** 4
**Rating:** 6
**Confidence:** 3

**Summary:**

This study highlights a previously overlooked yet strong and prevalent latent interaction rhythm pattern. It introduces a model called TG-Mixer model to capture this pattern, achieving superior performance in temporal link prediction.

First, the authors conduct empirical analyses to confirm that such latent interaction rhythm patterns exist in various real-world temporal graphs. They show, at both macro and micro levels, that real-world temporal graph interactions exhibit short-term interaction bursts compared to random interactions. This, in turn, creates a significant clustering pattern in node interaction rhythms.

Based on this finding, the authors propose TG-Mixer, a model to capture these clustering patterns. TG-Mixer samples a node's most recent historical links at each time point and encodes them using node and edge features and time intervals. The model also incorporates a "silent decay mechanism" to penalize long-term inactive nodes, thereby capturing temporal clustering patterns with a decaying approach. Through an additional LSTM-based information mixer, TG-Mixer combines temporal and structural information, along with temporal clustering signals, to form node representations, which are then used for link prediction.
Experimental results demonstrate that TG-Mixer outperforms other baselines regarding faster convergence, stronger generalization capabilities, and higher efficiency.

**Strengths:**

•	S1. Validity of Empirical Analysis: The authors hypothesize that temporal clustering patterns exist in real-world temporal graphs and verify this through empirical analysis. They introduce a "silence duration" metric to measure the degree of interaction bursts per node, comparing interactions in real-world temporal graphs to random pairs at both macro and micro levels. They conclusively show that real-world temporal graphs exhibit significantly more short-term bursty clustering patterns, making this approach and result convincing and significant.

•	S2. Originality: Unlike existing methods, this study introduces a novel approach to encoding temporal clustering patterns and demonstrates experimentally that these patterns play a significant role.

•	S3. Clarity: The research process is presented clearly. The authors empirically demonstrate the temporal clustering patterns overlooked by existing methods, explain how TG-Mixer encodes these patterns, and clearly show how it performs well in temporal link prediction. This structure makes the study easy to follow.

**Weaknesses:**

•	W1. Similarity of Techniques to Existing Methods: The stage of TG-Mixer in lines 259-271 of section 4 is similar to that of GraphMixer. Specifically, sampling the most recent historical links and encoding and padding historical link information closely resemble the link-encoder process in GraphMixer, even though the TG-Mixer encoder after that stage is a novel approach.

•	W2. Insufficient Detail in Method: The silent decay mechanism is a key component of this study, yet the explanation of the communal rhythm vector in lines 287-301 is insufficient. While additional details are provided in appendix A in the context of the Hawkes Process, the role of the rhythm vector is not clearly explained in the main text, making it difficult to understand its purpose.

**Questions:**

•	Q1. I want to provide a more detailed explanation of the rhythm vector in lines 287-301 of section 4. Is this rhythm vector updated globally whenever there is an interaction?

o	Q1.1 If the rhythm vector is indeed updated globally with each interaction, how does this work when TG-Mixer processes interactions in batches?

o	Q1.2 Additionally, how is the rhythm vector updated when multiple interactions occur simultaneously in a single time step? This may not matter at high-time resolutions, but I’m curious about the handling of multiple interactions at the same time.


•	Q2. In section 4, the method presents sampling the most recent historical links as a novel approach. However, I’m curious about the differences between this sampling method and the one used in GraphMixer. In lines 265-266, it is mentioned that GraphMixer samples a "massive number of selected historical neighbors." Still, I understand that GraphMixer also samples only the top K links when encoding edge information. Is there a difference here?

•	Q3. In many results of section 5, temporal link prediction performances in many TGNN models, including TG-Mixer, are evaluated using random negative samples. However, I’m curious whether TG-Mixer would still outperform other baseline models if evaluated using a more advanced negative sampling approach based on node degree distribution (i.e., the frequency of prior interactions).

•	Q4. In many experiments of section 5, TG-Mixer uses sample size as a hyperparameter, adjusting it differently for each dataset. How did you determine the optimal sample size for each dataset? Did you tune it on the validation set, or did you test multiple sample sizes on the test set and report the best performance?

---

> ### Author Response · Authors · 2024-11-20
> **Response to Reviewer S1Kp [Part 1]**
>
> We would like to sincerely thank Reviewer S1Kp for providing a detailed review with insightful questions.
>
> > **W1:** Reaffirmation of novelty and contribution.
>
> *Quick Answer: We present the **first attempt to investigate temporal clustering** in temporal graphs and successfully incorporate this insight for advancing temporal link prediction.*
>
> TG-Mixer extracts neighborhood information through a standard technique used in existing TGNs, but it is not the main focus of our work. As noted in Lines 75-107, we highlight our key contributions as follows:
>
> * **Empirical analyses for temporal clustering.** We introduce statistical empirical analyses and present the first attempt to identify temporal clustering in temporal graphs, i.e., the activity of certain nodes exhibits burstiness.
> * **A novel model for temporal link prediction.** Building on the observations from empirical analyses, we re-think the interaction dynamics and propose a novel model that explicitly captures burstiness information for temporal link prediction.
>
> > **W2 (Q1-1):** Motivation and update process of $\mathbf{C}_ \text{rhythm}^t$.
>
> *Revision: As shown in Lines 302-347 and 912-915, we have revised the paper to provide more details of the rhythm vector.*
>
> **Motivation Analysis.** As mentioned in Lines 303-310, we introduce the rhythm vector $\mathbf{C}_\text{rhythm}^t$ to encapsulate the condensed essence of interaction rhythms at each timestamp. This vector is updated globally and chronologically through the temporal mixer. Intuitively, the interaction rhythm for the next timestamp is influenced not only **by the current rhythm patterns (achieved by the silence decay mechanism)** but also **by the historical rhythms among recent interactions (achieved by the information mixer).**
>
> **Update Process.** Based on this motivation, **as clearly shown in the "Temporal Mixer" part of Figure 4**, the update process of the rhythm vector consists of two steps: (i) a silence decay mechanism that captures current rhythms through inter-event times (Eq. 3, 4), and (ii) the information mixer, which updates the rhythm vector with historical rhythms among recent interactions (Eq. 5). As revised in Lines 912-915, for batch processing, we maintain a matrix with a size corresponding to the batch size, which is dynamically updated throughout training.
>
> > **Q1-2:** Handling simultaneous interactions.
>
> *Quick Answer: We would like to clarify that as noted in Lines 1407–1414, we have already discussed this situation where TG-Mixer handles a large number of simultaneous interactions.*
>
> **Empirical Analysis.** We introduce interaction density to quantify simultaneous interactions among datasets. As mentioned in Line 1408 of Section D.4, we define interaction density (the averaged node degree at each timestamp) to quantify the degree of simultaneous interactions. When there is a large number of simultaneous interactions, it reflects high interaction density and more pronounced burstiness in nodes' activity, thus amplifying temporal clustering. Therefore, as demonstrated in Figure 6 in Line 448, the decay coefficient in the update process of the rhythm vector (Eq. 4) can leverage the shorter inter-event times to provide more distinguishable signals, leading to improved model performance.
>
> **Experimental Evaluation.** The results in Line 324 of Table 1 show that TG-Mixer **significantly outperforms baselines** on datasets with more simultaneous interactions, particularly on the US Legis. and Flights datasets. This may be because baselines struggle to provide differentiated encoding when multiple interactions share the same timestamp. In contrast, TG-Mixer effectively captures high-level temporal clustering, making it less susceptible to this issue. Further details can be found in Section D.4.
>
> > **Q2:** Discussion between TG-Mixer and GraphMixer.
>
> First of all, we would like to clarify that we did **not** state that GraphMixer samples a "massive number of historical neighbors". We referenced GraphMixer in this context to highlight the limitations of existing works, as these limitations are also supported by the GraphMixer.
>
> **Kindly note that our novelty and contribution extend far beyond the neighbor selection strategy (see W1 for further details).** While the neighbor selection strategy is technically similar to that of GraphMixer, the key difference lies in the motivation:
>
> * GraphMixer samples recent historical links with the goal of "avoiding poor model trainability and generalization ability".
> * In contrast, as mentioned in Lines 246-251, the neighbor selection strategy in TG-Mixer is driven by empirical analyses, where we believe that the latest rhythm patterns are critical for predicting future links.

---

> > ### Author Response · Authors · 2024-11-20
> > **Response to Reviewer S1Kp [Part 2]**
> >
> > > **Q3:** Evaluation under various negative sample strategies.
> >
> > *Quick Answer: TG-Mixer **demonstrates outstanding performance across four challenging negative sampling strategies** under the MRR metric, reinforcing its effectiveness.*
> >
> > *Revision: As revised in Lines 972-1007 of Section C.2, we have incorporated discussions of different negative sampling strategies.*
> >
> > **Empirical Analysis.** As stated in Line 908, We evaluate model performance using the random sampling strategy. Recently, some works [1, 2] have introduced advanced evaluation methods for temporal link prediction in temporal graphs. To fully assess the models, we adopt a more challenging evaluation with four negative sampling strategies and rank-based metrics.
> >
> > **Experimental Evaluation.** Based on the above motivation, we conduct additional experiments to evaluate the model performance under random negative sampling, historical negative sampling, inductive negative sampling, and degree-aware negative sampling using the MRR metric. For clarity, we first summarize our findings and analyses, followed by the results and implementation details. Other results and findings are provided in Line 972.
> >
> > From the results, we find that:
> >
> > * TG-Mixer **consistently demonstrates strong performance** across all negative sampling strategies under the MRR metric, further proving its robustness and effectiveness for temporal link prediction.
> > * All models experience some degree of performance degradation under all negative sampling strategies. Moreover, the performance of different models is more discriminative in these evaluation scenarios.
> > * Inductive negative sampling tends to cause the most performance drop. This is likely because this strategy samples unseen nodes to construct negative links, making it more challenging for models to accurately distinguish between positive and negative links.
> >
> > Table 1: Transductive MRR results under different negative sampling strategies on Reddit.
> >
> > ||JODIE|TGAT|TGN|GraphMixer|DyGFormer|TG-Mixer|
> > |:-:|:-:|:-:|:-:|:-:|:-:|:-:|
> > |random|70.24|64.24|72.95|70.38|81.47|**88.10**|
> > |historical|61.83|60.77|62.81|69.16|70.16| **73.80**|
> > |inductive|63.18|55.18|57.94|66.19|68.18| **70.51**|
> > |degree-aware|68.94|59.17|60.19|67.91|71.92|**74.24**|
> >
> > **Implementation Details.** In addition to the **random negative sampling** used in our main paper, inspired by [1], we employ **historical negative sampling** (sampling negative links that have been observed before) and **inductive negative sampling** (sampling negative links are not observed during training). To mitigate potential biases from popular nodes, we construct **degree-aware negative sampling** by sampling negative nodes with the probability of $d/N$, where $d$ is the node's current degree and $N$ denotes the total number of historical interactions. For the evaluation metrics, we follow [2] and sample 100 negative edges per positive edge and employ Mean Reciprocal Rank (MRR).
> >
> > [1] Towards Better Evaluation for Dynamic Link Prediction, NeurIPS, 2022.
> >
> > [2] Temporal Graph Benchmark for Machine Learning on Temporal Graphs, NeurIPS, 2023.
> >
> > > **Q4:** Explanation of different sample size $m$.
> >
> > *Quick Answer: Based on the model performance under different sample sizes in Tables 4 & 13, we find that it is necessary to tune the sample size based on the specific dataset.*
> >
> > **Model Configuration.** Kindly note that in Line 926 of our paper, we report the default sample size for the experimental results, even if we have investigated the sensitivity of the sample size.
> >
> > **Experimental Evaluation.** As shown in Tables 4 (Line 502) & 13 (Line 1192), we analyze the impact of different sample sizes $m$. The experimental results demonstrate that model performance exhibits varying sensitivities to the sample size across different datasets. Consequently, as noted in Line 499, we emphasize that tuning the sample size $m$ in different datasets is essential.
> >
> > Once again, thank you for your valuable review. You truly have helped us to improve our work.

---

> > ### Comment · Reviewer_S1Kp · 2024-11-25
> >
> > I really appreciate your valuable response and revision. Thanks to your responses, my concerns have been generally addressed. I agree with the contribution of this study, which lies in the empirical analysis of temporal clustering and its application to temporal link prediction. Based on this, I would like to raise the contribution score to 4 while maintaining the previous positive score. Thank you once again.

---

> ### Author Response · Authors · 2024-11-26
> **Thanks for your positive feedback and raising the contribution score!**
>
> Thank you once again for your constructive comments and for adjusting your contribution score during the discussion phase! Your positive feedback, **particularly regarding our novelty, contribution, and presentation**, has been incredibly inspiring to us. We are proud to present the first attempt at investigating temporal clustering in temporal graphs and leveraging this insight for temporal link prediction. Your detailed suggestions have been immensely valuable in enhancing our work.
>
> Once again, thank you for your dedicated efforts and the positive feedback!

---

### Author Response · Authors · 2024-12-04
**Thanks to Reviewers and Summary of Revisions**

Dear Reviewers,

We are deeply grateful for your insightful feedback and valuable suggestions. Your comprehensive reviews have guided us in making significant enhancements to our work.

**We present the first attempt to explore bursty patterns in temporal graphs and propose a novel model that leverages this insight to enhance temporal link prediction.** Firstly, we introduce empirical analyses to validate the existence of burstiness (we refer to as temporal clustering) in real-world temporal graphs. These findings lead to two key insights for predicting future links: the recent historical links and the inter-event times.
Therefore, different from existing works that explore various complex model architectures, we propose TG-Mixer, an effective and efficient method that explicitly considers interaction burstiness by (i) sampling the most recent historical links to extract neighborhoods; and (ii) penalizing the nodes’ long-term inactivity to incorporate temporal clustering information. TG-Mixer achieves SOTA performance on seven datasets from the DyGLib benchmark and five datasets from the TGB benchmark.

The reviewers have expressed positive feedback after reading our rebuttals, and some have raised their scores. We would like to express our appreciation for the positive recognition of the strengths of our work, including:

* Novelty and Originality. The introduced empirical analyses are **"novel and convincing"** (Reviewers S1Kp, qT7Y, AAW9), and the proposed method demonstrates **"strong originality"** (Reviewers S1Kp, nfv5, qT7Y, AAW9, TU2Y).
* Strong Motivation. The overall framework is **"well-motivated"** (Reviwers S1Kp, qT7Y, AAW9), and all components are **"well-designed and sound reasonable"** (Reviwer S1Kp).
* Comprehensive Evaluation. Extensive experimental results demonstrate **"reproducibility"** (Reviewer AAW9) and are **"persuasive enough"** to show the method's effectiveness and efficiency (Reviewers S1Kp, nfv5, qT7Y, AAW9, TU2Y).
* Clear Presentation. The paper is **"well-written"** (Reviewers S1Kp, qT7Y, AAW9) and **"easy to follow"** (Reviewers S1Kp, AAW9).

We have responded individually to each reviewer's questions. To address your concerns and enhance our submission, we have incorporated your suggestions by providing additional clarifications and conducting further experiments. See our revised paper and Appendix for details. Below is a summary of the key updates:

* **Toy illustrative example (Reviewer TU2Y).** We have included a toy example to effectively demonstrate the temporal clustering patterns we introduced. This example provides clear interpretations, making our paper more accessible and easier to understand.
* **Expanded related work (Reviewers nfv5, qT7Y).** We have expanded the Related Work section to include relevant research on burstiness. The comparative discussion between these studies and ours highlights the novelty and contributions of our work.
* **Detailed description of the model component (Reviewers S1Kp, nfv5, TU2Y).** For better readability, we have elaborated on the description of the TG-Mixer encoder module. This includes relevant formulas, detailed comprehensive explanations, and the motivation and purpose behind each stage.
* **Experiments for ablation study (Reviewer qT7Y).** We have conducted an additional ablation study on the silence decay mechanism. The observed performance drop when removing this mechanism demonstrates its effectiveness and the importance of temporal clustering information.
* **Experiments on the TGB benchmark (Reviewers nfv5, AAW9).** We have conducted further evaluations using additional datasets from the TGB benchmark. TG-Mixer continues to achieve the best performance across these additional datasets, further confirming its superiority.
* **Experiments under four negative sampling strategies (Reviewers S1Kp, nfv5, qT7Y, AAW9).** We have performed additional evaluations using four negative sampling strategies: random, historical, inductive, and degree-aware sampling. Under these extensive comparisons, TG-Mixer still delivers the best performance.
* **Experiments under different batch sizes (Reviewer nfv5).** We have evaluated the model performance under various batch sizes. TG-Mixer consistently exhibits strong performance, underscoring its robustness. Furthermore, based on our empirical results, we identify a fundamental limitation in the widely used batch-based training approach of existing methods, which warrants further investigation.
* In addition to the above revision, we plan to include an additional ablation study on our batch-based rhythm vector and the experimental results for MRR with fixed source nodes (Reviewer AAW9).

Once again, we sincerely thank all reviewers for your valuable feedback and constructive opinions, which have significantly enhanced our work.

Best,
Submission 5734 Authors

---

### Meta-Review · Area_Chair_VSZ3 · 2024-12-20

**Metareview:**

The authors propose a novel Temporal Graph Neural Network (TGN) inspired by burstiness patterns observed in real-world interaction datasets. Specifically, based on empirical patterns, they introduce a mechanism to sample recently occurring interactions and penalize long-term inactive nodes within the TGN framework. Extensive experiments demonstrate the effectiveness of the proposed approach.

Most reviewers acknowledged the following strengths:
- S1. The novelty of the proposed TGN architecture
- S2. Its strong empirical performance
- S3. The comprehensiveness of the experiments conducted

On the negative side, the reviewers raised the following concerns:
- W1. While the authors claim their empirical observations to be novel, there is already *substantial* literature on similar observations
- W2. The proposed method has an undesirable dependence on training batch sizes, resulting in a misalignment with the problem definition and introducing complexity that can lead to inefficiency
- W3. The presentation, particularly the explanation of method details, requires improvement

The paper received mixed reviews, with no reviewer expressing strong enthusiasm for its acceptance. Moreover, the meta-reviewer identified W1 as a significant concern, and the overlook of closely related studies may have positively influenced the evaluation of the reviewers who were unaware of them.

Considering these factors, the meta-reviewer recommends rejecting the paper and encourages the authors to address the concerns raised (particularly W1 by providing detailed comparisons and clearly specifying their contributions).

**Additional Comments On Reviewer Discussion:**

There was extensive discussion between the authors and the reviewers, but the reviewers found their concerns partially unaddressed.

---

### Decision · Program_Chairs · 2025-01-22

Reject